# Cross-Channel Activation Function with Pass-Through Ratio Control

## Abstract

In convolutional neural networks (CNNs), activation layers process features from convolutional layers, which have multiple output channels. Conventional activation functions like ReLU handle these multi-channel features independently, ignoring spatial and cross-channel dependencies. This hard-thresholding approach can lead to information loss by eliminating negative features and disrupting the connection within input features. To address this issue, we propose a novel activation function that considers mutual relations across multiple channels. Our activation layer processes tuples across channels as single inputs, ensuring that output tuples remain in the same projection space, with their $\ell_1$ norms bounded by a learnable parameter. This parameter controls the pass-through ratio, which is the proportion of input data allowed to pass through the activation layer, offering a significant advantage over ReLU. Our approach demonstrated superior accuracy in classification tasks on common benchmarks and domain-specific datasets for CNN-based models. The proposed activation layer outperformed ReLU and other common layers in both clean and noisy data scenarios, as confirmed by statistical tests. Our results highlight the effectiveness of this activation function in maintaining feature integrity and improving model performance.

## 1 Introduction

Neural networks (NNs) are nonlinear functions that map inputs to outputs through layers performing operations like convolution, pooling, and activation. Each layer can be represented as $y = \phi(Wx + b)$, where $W$ and $b$ are weights and biases, and $\phi$ is an activation function. Activation layers are crucial for capturing nonlinearity and sparsity in data, with a suitable choice enhancing network performance, stability, and noise robustness. While many nonlinear activation functions exist, finding the optimal one involves trade-offs due to conflicting desirable properties. The Rectified Linear Unit (ReLU) is popular for its simplicity but suffers from the *dying ReLU problem* [Lu et al. (2020)], where neurons can become inactive.

To overcome ReLU's limitations, various ReLU-like functions, such as Leaky ReLU [Maas et al. (2013)], Parametric ReLU (PReLU) [He et al. (2015)], and GELU [Hendrycks & Gimpel (2016)], have been developed to retain advantages while addressing drawbacks. Despite their effectiveness [Szandała (2021)], ongoing research into better activation functions is necessary, employing strategies like genetic algorithms [Basirat et al. (2019)] and learning-based approaches [Ramachandran et al. (2018)]. Adaptive activation functions, which learn parameters during training, represent the most advanced development in this area.

Existing ReLU-like functions face limitations, particularly their element-wise application and lack of trainable parameters, preventing them from fully utilizing relationships in input data. This is especially important in convolutional neural networks (CNNs), where multiple output channels must be considered. These functions often process inputs separately, neglecting dependence between them, such as the spatial or cross-channel relation of the features. Spatial relation refers to the local connectivity and neighborhood structure of the features, while cross-channel relation refers to the correlation and diversity of the features across different channels. These relations are important for capturing the patterns and semantics of the input data, and features extracted in the previous layers of the network. Moreover, ReLU-like activation may lose the connection with the input features

and cause information loss due to fixed-threshold eliminating negative features in an element-wise manner.

To address these issues, we propose the Simplex Projection Activation (SPA), a cross-channel activation function that maintains feature relations and connection between input and output. See illustration of SPA in Fig. 1a. SPA was shown to improve classification accuracy on CNN models across multiple datasets, including noisy data, and is a strong alternative to ReLU.

## 2 RELATED WORK

In this section, we review related works on activation functions, and highlight the novelty and advantages of our proposed simplex projection activation (SPA) function.

**Channel-wise activation functions** use information from several elements of the input data, rather than applying a pointwise transformation to each element. Examples include the Maxout unit [Goodfellow et al. (2013)], which selects the maximum value among several linear transformations, and meta-ACON functions [Ma et al. (2021)], which incorporate layer-wise and channel-wise adaptive parameters. The Variable Activation Function (VAF) [Apicella et al. (2019)] uses affine transformations before and after activation to capture cross-channel dependencies, enhancing network performance. Additionally, the study on adaptive activation functions [Liu et al. (2020)] introduced parameterized S-shaped and ReLU-like functions that dynamically adjust during training, improving accuracy across tasks. These approaches demonstrate the importance of parameterized and channel-wise activations in modern neural network design.

**Concept-based activation functions** derive from principles that guide network learning or inference. Stochastic activation functions [Urban et al. (2017); Shridhar et al. (2019); Chen et al. (2019)] introduce randomness to outputs, while the ACON family [Ma et al. (2021)] extends Maxout with adaptive Swish-like functions. Lifted Neural Networks [Askari et al. (2018); Sambharya (2018)] frame activation as solutions to optimization problems, replacing non-smooth functions with smooth penalties. Furthermore, the introduction of Deep Sparse Rectifier Networks [Glorot et al. (2011)] demonstrated how sparsity in activations benefits neural network performance. By employing the ReLU activation, these networks achieved significant improvements in training efficiency. However, the limitations of ReLU, such as the *dying ReLU problem*, highlight the need for alternative methods to maintain active neurons during training.

**Gaussianization and normalization transformations** are relevant for preprocessing and feature transformation. The Generalized Divisive Normalization (GDN) [Ballé et al. (2015)] introduced a parametric nonlinear transformation to Gaussianize data from natural images. GDN reduces mutual information between components by combining a linear transformation with divisive normalization. This approach demonstrates how decorrelation improves density modeling and feature distribution. Inspired by these principles, simplex projection techniques incorporate similar constraints, ensuring well-regularized feature spaces.

**Simplex projection applications** utilize simplex projection in output or intermediate layers of neural networks, serving as an alternative to the softmax layer [Askari et al. (2018)] and producing probability distributions without the associated numerical instability. Convolution Simplex Projection Networks [Briq et al. (2018)] integrate simplex projection into CNNs to improve segmentation heatmap quality and incorporate additional loss terms.

However, these applications do not explore simplex projection as an activation function in hidden layers. Unlike most activation functions that act on individual input elements, SPA accounts for feature dependence across channels, projecting input tuples onto a convex set that preserves their mutual relations and avoids information loss. This generalization of the ReLU function enhances information retention, sparsity, regularization, robustness, and overall network performance.

## 3 CROSS-CHANNEL ACTIVATION FUNCTION

### 3.1 ACTIVATION LAYER AS AN OPTIMIZATION PROBLEM SOLVER

We reinterpret the ReLU activation as an optimization problem. For an input feature $\mathbf{X}$, which usually has the shape of $C$ (channels) $\times H$ (height) $\times W$ (width), the ReLU layer returns the

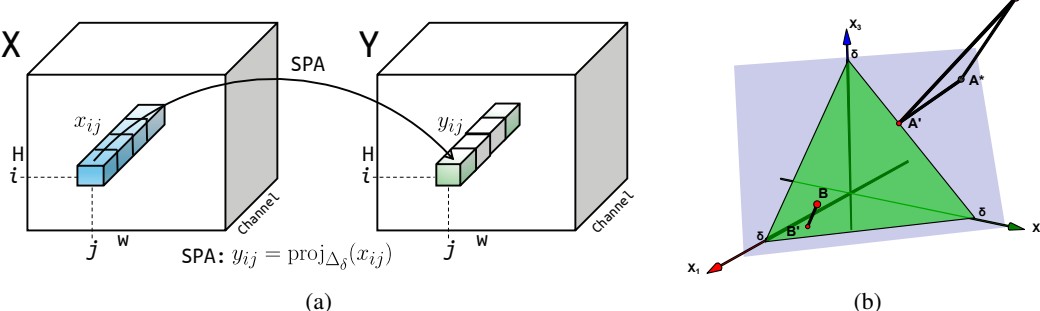

(a)                                                                    (b)

Figure 1: The simplex projection illustration. (a) SPA projects an input tuple $\boldsymbol{x}_{ij}$ to an output tuple $\boldsymbol{y}_{ij}$ onto the $\delta$ simplex across all channels within a multi-channel input $\mathbf{X}$. Null elements are shown in white. (b) Illustration of $\delta$-simplex projection for a 3-dimensional case. Point $A = [-\delta, 0.8\,\delta, \delta]$, with negative coefficients, is projected to $A' = [0, 0.4\,\delta, 0.6\,\delta]$ on the simplex border. Point $B$ is projected inside the simplex.

elements of the output individually as $y = \max(x, 0)$. This is equivalent to projecting each element $x = \mathbf{X}(c, i, j)$ onto the nonnegative orthant, i.e., solving the optimization problem:

$$y = \arg\min_{y} \quad |y - x|^2, \tag{1}$$

$$\text{s.t.} \quad y \geq 0, \tag{2}$$

This interpretation has been introduced in [Agrawal et al. (2019)], and brings some new insights to extend the ReLU layer. We note that elements of the input features, $\mathbf{X}(c, i, j)$, are often not completely statistically independent. Features in the some first layers of the neural networks exhibit high dependence, while treating them individually may disrupt their latent connection, e.g., changing their covariance matrices, or feature lengths especially when data is corrupted by noise.

## 3.2 SIMPLEX PROJECTION ACTIVATION

To preserve the feature dependence and avoid information loss, we propose a new type of activation function that considers the feature mutual relation across multi-channels. We define $\mathbf{X}(:, i, j)$ as a tuple of $C$ feature elements of $\mathbf{X}$ at the same location $(i, j)$. Instead of projecting each individual element $\boldsymbol{x}$ of $\mathbf{X}$ onto the nonnegative orthant, we project each tuple $\boldsymbol{x} = [x_1, x_2, \ldots, x_C]$ of $\mathbf{X}$ onto a convex set $S$. Here, $C$ is the number of channels, and $S$ imposes non-negativity and bound constraints on the mutual relations among features. For example, $S$ can be defined as the set of tuples whose $\ell_1$-norms are equal to or bounded above by a constant $\delta$, i.e.,

$$S = \{x = [x_1, x_2, \ldots, x_C] \,|\, x \geq 0, \|x\|_1 \leq \delta\}.$$

This approach enforces sparsity and regularization on the output tuples, limits their magnitude, and ensures that the outputs remain in the same projection space, preserving their mutual relations. The proposed activation function can be formulated as

$$\boldsymbol{y} = \arg\min_{\boldsymbol{y}} \quad \frac{1}{2}\|\boldsymbol{y} - \boldsymbol{x}\|_2^2, \tag{3}$$

$$\text{s.t.} \quad \boldsymbol{y} \in S. \tag{4}$$

The $\ell_1$-norm inequality constraints can be replaced by equality constraints, e.g., by introducing a dummy variable $z \geq 0$ such that $z + \mathbf{1}^T \boldsymbol{y} = \delta$. The projection can be reformulated as a projection onto the $\delta$-simplex, that is

$$\boldsymbol{y} = \arg\min_{\boldsymbol{y} \in \mathbb{R}^C} \quad \frac{1}{2}\|\boldsymbol{y} - \boldsymbol{x}\|_2^2, \tag{5}$$

$$\text{s.t.} \quad \boldsymbol{y} \geq 0, \quad \mathbf{1}^T \boldsymbol{y} = \delta, \tag{6}$$

where $\delta > 0$ is a learnable parameter of the layer, and $\mathbf{1}$ is the vector of ones. In this paper, we consider the projection in (5). We call this activation function *Simplex Projection Activation* (SPA). See Fig. 1a for an illustration of SPA.

### 3.2.1 Optimal $\delta$-simplex projection

Assume that elements of the tuple $\boldsymbol{x}$ are sorted in the descending order, i.e., $x_1 \geq x_2 \geq \cdots \geq x_C$. The projection of $\boldsymbol{x}$ onto the probability simplex has been studied in [Boyd & Vandenberghe (2004); Chen & Ye (2011)]. For the problem in (5), the Lagrangian function and its gradient w.r.t to $\boldsymbol{y}$ are given by

$$L(\boldsymbol{y}, \boldsymbol{\lambda}, \nu) = \frac{1}{2}\|\boldsymbol{x} - \boldsymbol{y}\|_2^2 - \nu(\mathbf{1}^T\boldsymbol{y} - \delta) - \boldsymbol{\lambda}^T\boldsymbol{y},$$

$$\nabla_y L = \boldsymbol{y} - \boldsymbol{x} - \mathbf{1}\nu - \boldsymbol{\lambda} = 0,$$

where $\boldsymbol{\lambda} = [\lambda_1, \ldots, \lambda_C] \geq 0$ and $\nu$ are the Lagrange multipliers associated to the inequality and equality constraints in (6). Setting the gradient $\nabla_y L$ to zero gives the optimal solution $\boldsymbol{y}^\star = \boldsymbol{x} + \mathbf{1}\nu + \boldsymbol{\lambda}$.

Let $\mathcal{I}$ be the index set of positive elements $y_i > 0$, $i \in \mathcal{I}$. Due to the KKT complementary slackness condition for the nonnegativity constraint, we have $\lambda_i = 0$, $i \in \mathcal{I}$, implying that $y_i^\star = x_i + \nu$. Considering the equality constraints $\sum_i y_i^\star = \sum_{i \in \mathcal{I}} y_i^\star = \delta$, we can derive the optimal dual

$$\nu^\star = \frac{1}{I}(\sum_{i \in \mathcal{I}} y_i^\star - x_i) = \frac{1}{I}(\delta - \sum_{i \in \mathcal{I}} x_i) = \frac{\delta}{I} - \bar{x}_{\mathcal{I}}, \tag{7}$$

where $I = |\mathcal{I}|$, $\bar{x}_{\mathcal{I}} = \frac{1}{I}\sum_{i \in \mathcal{I}} x_i$ is the mean of $x_{i \in \mathcal{I}}$. In addition, $y_i^\star = x_i + \nu^\star > 0$ for all $i \in \mathcal{I}$ implies that $\mathcal{I}$ is the index set of all $x_i > -\nu^\star$, i.e., $\mathcal{I} = \{i : x_i > -\nu^\star\}$. Obviously, for zero elements $y_{j \notin \mathcal{I}} = 0 = x_j + \nu^\star + \lambda_j \geq x_j + \nu^\star$, since $\lambda_j \geq 0$. It means $-\nu^\star$ is the midpoint which splits the tuple $\boldsymbol{x}$ into two disjoint sets

$$x_{i \in \mathcal{I}} > -\nu^\star \geq x_{j \notin \mathcal{I}},$$

or

$$x_1 \geq \cdots \geq x_I > -\nu^\star \geq x_{I+1} \geq \cdots \geq x_C.$$

This suggests an algorithm to determine the largest $I$ elements $\{x_1, \ldots, x_I\}$ such that the smallest element $x_I > \bar{x}_{\mathcal{I}} - \frac{\delta}{I}$ or equivalent condition

$$\delta > \sum_{i=1}^{I-1}(x_i - x_I). \tag{8}$$

Since the elements $x_i$ are sorted in the descending order, $x_i - x_I \geq 0$ for all $i = 1, \ldots, I$. The set $\mathcal{I}$ always contains at least the largest element $x_1$.

The final output $y_i = x_i + \nu^\star$ for all $i \leq I$, and $y_i = 0$ for all $i > I$. Thus, the update rule for each tuple $\boldsymbol{x}$ is

$$\boldsymbol{y} = \max(\boldsymbol{x} + \nu^\star, 0). \tag{9}$$

**Remark.** *The formulation of the SPA layer as a convex optimization problem (projection of feature vectors onto the probability simplex) ensures a globally optimal solution, derived through the proposed efficient update rule without requiring iterative algorithms.*

**Remark.** *Different from ReLU, SPA shifts the input, $\boldsymbol{x}$, by $\nu^\star$ in (7), i.e., centered by $\bar{x}_{\mathcal{I}}$ then shifted by $\frac{\delta}{I}$ before nullifying negative elements. The SPA function tends to pass more features in the early layers than ReLU, and suppresses more input features to zero in the final layer.*

**Remark.** *When all elements of a tuple are negative, ReLU returns a tuple of zeros, which means that it discards all the information from the input tuple. This can cause information loss and reduce the network's ability to learn from the data. The SPA in (9) returns a tuple that has at least one non-zero element, the largest element in the input tuple. This means that it preserves the information from the input tuple, and assigns the highest probability to the most relevant feature. This can enhance the information retention and improve the network's ability to learn from the data.*

**Remark.** *The SPA activation focuses on the cross-channel dependencies within a feature tuple at each spatial location, grouping channels identified as relevant and nullifying the less significant ones. Convolutional operations, in turn, are primarily designed to learn spatial local information by applying shared kernels over small receptive fields, capturing relationships within the neighborhood of spatial locations. This distinction reflects the separation of spatial learning (via convolution) and channel-level feature selection (via activation).*

SPA has several advantages over ReLU and its variants.

- SPA can capture the cross-channel feature dependence and avoid information loss due to the hard-thresholding rectifier.

- SPA can nullify (sparsify) group of multiple features simultaneously. The features in the later layers of the neural network may have some redundant or irrelevant elements that do not contribute to the task. Applying the ReLU activation individually to each element may keep some of these elements, and increase the network complexity and overfitting. SPA, on the other hand, can eliminate some of these elements together.

- Projection on the probability space (simplex) and sparsification of the features also implies the features on more important channels are preserved, whereas the output tuples have a probabilistic interpretation.

- SPA also improves the network robustness by constraints on bound of the activation outcomes. SPA can normalize or adapt the features to a suitable scale or range ($\delta$), is able to improve the network stability.

### 3.2.2 PASS-THROUGH RATIO BY THE PARAMETER $\delta$

The parameter $\delta$ of the SPA layer in (3) controls the scales of the simplex and can be considered as a learnable parameter or a hyperparameter. The scale of the simplex, $\delta$, controls how much the input tuples are normalized and sparsified by the SPA function, thereby controlling the *Pass-Through Ratio* (PTR), which is the proportion of input data allowed to pass through the activation layer. From (8) and the final update rule (9):

- **Small $\delta$:** A small $\delta$ results in output tuples with a small sum, hence a high sparsity. SPA eliminates most of the small positive features and highlights the most relevant features by normalizing their sum to $\delta$. However, this may lead to over-sparsification, which can discard some useful information and reduce the network performance and convergence. For instance, when $0 < \delta < x_1 - x_2$, given that $x_1$ and $x_2$ are distinct, SPA returns the outcome $[\delta, 0, \ldots, 0]$ with only one non-zero element.

- **Large $\delta$:** A large $\delta$ results in output tuples with a large sum and low sparsity, which can retain most of the input features and avoid information loss. However, this may decrease the non-linearity properties of SPA, thereby decreasing the abilities of the neural network to learn complex patterns. If $\delta$ is too large, all of the input data will be projected inside a $\delta$-simplex, and the SPA layer will degenerate into a linear transformation. For example, when $\delta > \sum_{i=1}^{C} x_i - Cx_C$, SPA bypasses all elements through the layer as $\boldsymbol{y} = \boldsymbol{x} - \frac{1}{C}\sum_i x_i + \frac{\delta}{C}$.

To understand the influence of the parameter $\delta$ on the pass-through ratio, the experiments with the Gaussian distributed inputs were conducted. For the considered case, if input of size belongs to $\mathcal{N}(\mu_{in}, \sigma_{in})$ the mean value of the output distribution, $\mu_{SPA} = k_\mu \cdot \delta/C$, where $k_\mu$ is linear coefficient, equal to 1 for normal distribution, $\delta$ is a parameter of the SPA layer, and $C$ is the number of channels. The pass-through ratio, $PTR$, can be defined as:

$$PTR_{SPA} = f_{PTR}(\delta/(C \cdot \sigma_{in})), \tag{10}$$

where $\delta$ is a parameter of the SPA layer, $C$ is the number of channels, $\sigma_{in}$ is a variance parameter of the input Gaussian distribution, and $f_{PTR}$ is a non-linear function. Thus, in order to hold the same value of the path-trough ratio, we should save the ratio $\delta/(C \cdot \sigma_{in})$ to be constant. The next note can be that the PTR and the mean value of the output distribution of the SPA layer do not depend on the mean value of the input distribution.

## 4 EXPERIMENTS

**Datasets.** We evaluated the performance of activation layers using a diverse set of datasets, including common benchmarks such as `MNIST`, `FashionMNIST`, `CIFAR10/100`, `Caltech256`, `Tiny ImageNet`, and `ImageNet`. Additionally, a smaller subset of `CIFAR10` (referred to as `CIFAR10-5K`) was created for comparison on smaller datasets. Domain-specific datasets such as

`GTSRB` (traffic signs), `SVHN` (street numbers), and seven biomedical datasets from `MedMNIST` were also tested. The description of the datasers is presented in Appendix B.

To assess robustness, experiments with noisy data were conducted by adding Gaussian noise at five different noise levels ($\sigma$ = 0.05, 0.1, 0.2, 0.3 and 0.4). The noisy data was formed once and was unchangeable. In other words, the same noisy samples were used across different trials and activation functions. Examples of noisy images (Figure 4) and additional details can be found in Appendix B.

**Models.** To test the main concept, the first experiments were conducted for 3-layer CNN model (hereinafter SmallCNN). To validate the proposed activation layer for deep neural networks, extensive experiments were conducted for VGG16 and ResNet-18 networks. More details of the used neural networks are presented in Appendix C.

**Training procedure.** For SmallCNN and VGG model (`CIFAR10/100` dataset), Adam optimiser [Kingma & Ba (2015)] with constant learning rate was used. For `ImageNet`, `Tiny ImageNet`, and `Caltech256` dataset, SGD optimizer was employed with learning rate schedule for VGG and ResNet-18 models. For `MedMNIST` datasets, the training parameters followed the setup in Yang et al. (2021). The detailed description of the training procedure is in Appendix C.

**Metric evaluation and comparison.** In addition to a comparison of SPA with ReLU, we conducted simulations for GELU activation. It is known that GELU is efficient and resistant to noisy data Hendrycks & Gimpel (2016). This comparison aims to investigate whether SPA can offer comparable or improved performance over GELU. However, the comparison with other activation layers, like PReLU He et al. (2015), ELU Clevert et al. (2016), and SELU Klambauer et al. (2017), were also performed and can be find in Appendixes D and E. For each trial, a model was trained during the corresponding number of epochs, and the best accuracy on the validation data was taken. All utilized datasets are balanced across the classes. Hence, accuracy was used as an evaluation metric. A permutation test based on Student t-test (100,000 permutation) was selected for statistical comparisons Yuen & Dixon (1973); Yuen (1974); Hemerik & Goeman (2018) of the accuracy results. This test is more robust to a non-normal distribution but retains good interpretation ability.

**Experiment procedure.**

For the comparison of activation layers, we employed the following methodology. Conducted multiple training runs (not less than 10) with different initial weights and random order of training samples within epochs for each activation function while maintaining similar training parameters. We then performed statistical tests to compare the proposed activation (e.g., SPA, GELU) with ReLU. Finally, we reported mean accuracy, standard deviation (in parentheses), differences in mean accuracies compared to ReLU ($\Delta$), and p-values. This methodology was applied consistently across all experiments with small and medium-scale datasets. For the `ImageNet-1k` dataset, fewer trials were conducted, and the reported results consist of the median and the range values of the accuracies.

In addition, to evaluate the average ability of the activation layer to resist noise in the data, we examined how the activation layer performed under various noises for several benchmark datasets. Ten independent versions of noisy data were created for different noise levels. For each noise version, the best accuracy (the best local minimum) from several trials (not less than 3) of model training was selected. This procedure emulates the practical case, where we can train a model several times to find the best local minimum. The further statistical comparisons were conducted according to the methodology described above.

**Hyperparameter $\delta$.** The initialization method of the parameter $\delta$ depends on the dataset and the used model. The detailed description of the proposed methods is presented in Appendix F. For small models, the optimal values of the $\delta$ parameters can be found by a simple search. This method was applied for experiments with SmallCNN. For bigger models, like VGG16, the $\delta$ can be initialized based on intuition, and the final values of the $\delta$ after training can be taken. Then, we can use these values for initialization of the next iteration. This approach showed good results for VGG16 and `CIFAR10/100` datasets. However, this approach is not applicable for more difficult datasets or for training with weigh decay, where the $\delta$ values can change dramatically and become very small at the end of the training.

For the relatively big datasets and deep neural networks, we suggest using the approach based on the similarity between the effects of the SPA and ReLU on the output distribution. For the normalized

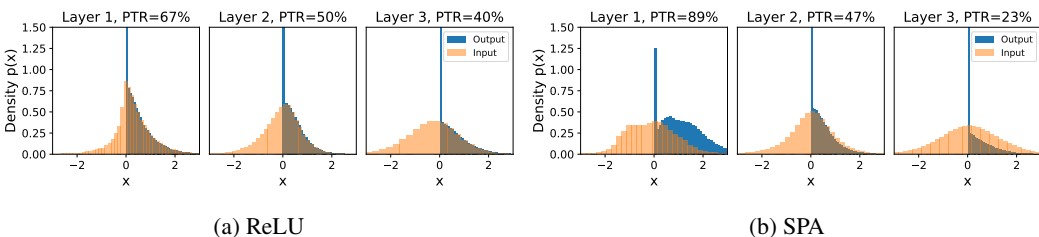

(a) ReLU                                          (b) SPA

Figure 2: Distribution of input/output data for different activation layers in SmallCNN. The distributions are presented for noise-free CIFAR10 dataset.

Gaussian input distribution $\mathcal{N}(0, 1)$, we can initialize $\delta$ as $0.4 \cdot C$, where $C$ is a number of channels. In this case, the form and mean value of the output distribution and pass-through ratio for SPA and ReLU actvations will be similar. This initialization method showed good results for `Tiny ImageNet` and `Caltech256` datasets. It should be noted that weight decay parameter should be different for $\delta$ values as it forces $\delta$ to decrease during training procedure.

In addition, Bayesian optimization (BO) Snoek et al. (2012) can be used to find the optimal delta values. For ResNet-18, there are 17 activation layers, and it will be very time-consuming to optimize all these parameters separately. To decrease the range of the BO search, we consider the finding multipliers of the $\delta$ values, which were initially set up based on the similarity between the effect of the SPA and ReLU on the output distribution. For this case, $\delta$ values are the same for the same number of channels, so we can reduce the number of considered parameters. This initialization method was used for `MedMNIST` datasets.

## 4.1 RESULTS

**Result for SmallCNN.**

The statistical comparison of the accuracy values for SmallCNN for different datasets is presented in Table 1. In this experiments, all three activation layers of SmallCNN were replaces on SPA with $\delta = 20$ or GELU. The only exception is for CIFAR100 dataset, where only two first activation layers were replaced by SPA. The results show the superiority of the proposed SPA layer for `MNIST`, `FashionMNIST` (FMNIST), and `CIFAR10`, `GTSRB` and `SVHN` datasets for noisy-free data. For `CIFAR10-5K` and `CIFAR100`, the accuracy of SPA was similar to ReLU.

In addition, SPA layer also showed superiority for noisy datasets. As example, the results for experiments with independent noise versions for `CIFAR` datasets are presented in Table 8 in Appendix D. SPA showed superiority for all noise levels for `CIFAR10` and for high noise levels for `CIFAR100` (for $\sigma = 0.3$ and 0.4) and `CIFAR10-5K` (for $\sigma = 0.1, 0.2$, and 0.3). The results for other datasets and other activation layers can be seen in Appendix D.

**Results of Deep models.** The results for the noise-free dataset using VGG16 and ResNet-18 are depicted in Table 1. An example of the results for noisy data is presented in Table 2.The full results can be found in Appendix E. A detailed description of the methods used for $\delta$ initialization is provided in Appendix F.

The results showed the superiority of the proposed SPA layer over ReLU (and GELU) for most tested datasets for both noise-free and noisy data. Specifically, SPA showed better accuracy for VGG16 on the `CIFAR10/100` and `Tiny ImageNet` datasets for noise-free data. For noisy data, superiority was observed on `CIFAR10/100` under all noise levels and `Tiny ImageNet` under low noise. For ResNet-18, SPA showed superiority on `Tiny ImageNet` and `Caltech256` datasets across all noise levels, including noise-free data. ResNet-18 also showed better accuracy for 6 out of 7 tested datasets for noise-free data. SPA showed results similar to ReLU only for `DermaMNIST`. However, for the noisy case, SPA showed superiority on this dataset and most other tested datasets from the `MedMNIST` database. The results for `ImageNet` (see Table 3 and Appendix E) showed that SPA achieves accuracy of 66.74% (range: 66.61%–66.85%) while ReLU has accuracy of 66.30% (range: 66.19%–66.57%). In other words, SPA achieves slightly higher accuracy than ReLU, with a non-overlapping range.

Table 1: Accuracy results for noise-free datasets with ReLU, GELU, and SPA activations.

| Dataset | ReLU | GELU | | | SPA | | |
|---|---|---|---|---|---|---|---|
| | Acc., % | Acc., % | Δ | p* | Acc., % | Δ | p* |
| SmallCNN | | | | | | | |
| MNIST | 99.48 (0.05) | 99.51 (0.04) | 0.03 | 0.0554 | **99.58 (0.04)** | **0.10** | **0.0000** |
| FMNIST | 91.88 (0.21) | 91.87 (0.17) | -0.01 | 0.8106 | **92.30 (0.10)** | **0.42** | **0.0000** |
| CIFAR10 | 84.43 (0.29) | **84.65 (0.20)** | **0.22** | **0.0072** | **84.86 (0.25)** | **0.42** | **0.0000** |
| CIFAR10-5K | 69.83 (0.59) | 70.17 (0.61) | 0.33 | 0.0867 | 70.11 (0.75) | 0.27 | 0.2066 |
| CIFAR100[1] | 57.46 (0.37) | 57.41 (0.34) | -0.06 | 0.5965 | 57.48 (0.41) | 0.02 | 0.8976 |
| GTSRB | 93.94 (0.44) | 93.74 (0.33) | -0.20 | 0.2512 | **95.51 (0.47)** | **1.56** | **0.0000** |
| SVHN | 90.47 (0.33) | *89.99 (0.30)* | *-0.47* | *0.0010* | **92.30 (0.20)** | **1.83** | **0.0000** |
| VGG16 | | | | | | | |
| CIFAR10 | 92.21 (0.16) | **92.38 (0.13)** | **0.17** | **0.0234** | 92.43 (0.14) | 0.22 | 0.0042 |
| CIFAR100 | 67.56 (0.17) | **68.07 (0.25)** | **0.51** | **0.0000** | **68.29 (0.18)** | **0.73** | **0.0000** |
| Tiny ImageNet | 50.00 (0.39) | **51.88 (0.38)** | **1.88** | **0.0000** | **52.66 (0.27)** | **2.66** | **0.0000** |
| ResNet-18 | | | | | | | |
| Tiny ImageNet | 53.01 (0.35) | **53.49 (0.48)** | **0.48** | **0.0164** | **54.31 (0.26)** | **1.30** | **0.0003** |
| Caltech256 | 67.70 (0.31) | *66.72 (0.18)* | *-0.98* | *0.0000* | **68.95 (0.38)** | **1.25** | **0.0000** |
| ResNet-18 (28)[2] | | | | | | | |
| PathMNIST | 85.39 (0.74) | 85.98 (0.73) | 0.59 | 0.0876 | **87.06 (0.59)** | **1.67** | **0.0000** |
| PneumoniaMNIST | 90.91 (0.78) | *90.00 (0.92)* | *-0.91* | *0.0271* | **91.75 (0.55)** | **0.84** | **0.0126** |
| BreastMNIST | 83.14 (3.71) | **86.86 (2.08)** | **3.72** | **0.0117** | **87.66 (1.35)** | **4.52** | **0.0007** |
| DermaMNIST | 75.16 (0.49) | 75.03 (0.41) | -0.13 | 0.2778 | 75.23 (0.37) | 0.07 | 0.5185 |
| OrganAMNIST[3] | 91.12 (0.33) | 90.98 (0.30) | -0.14 | 0.3228 | **91.55 (0.23)** | **0.42** | **0.0043** |
| OrganCMNIST[4] | 90.27 (0.22) | **90.48 (0.17)** | **0.20** | **0.0156** | **90.68 (0.23)** | **0.41** | **0.0001** |
| OrganSMNIST[3] | 76.99 (0.37) | 76.97 (0.30) | -0.01 | 0.8967 | **77.27 (0.41)** | **0.29** | **0.0240** |

**Note:** * p < 0.05 are marked in bold for higher accuracy and in italic for lower accuracy.[1] only first two layers of SmallCNN were replaced. [2] input image size of $28 \times 28$. [3,4] $\delta$ were initialized from the main and generalized setups based on Bayesian Optimization, accordingly.

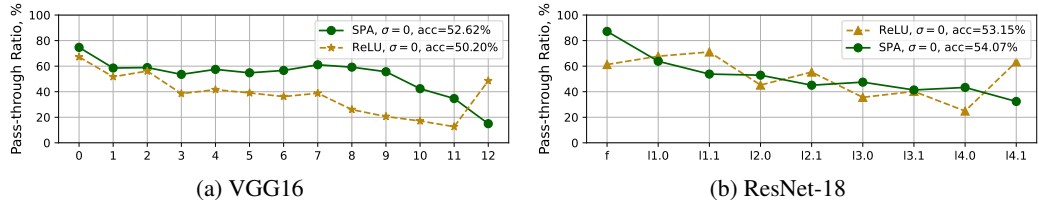

(a) VGG16      (b) ResNet-18

Figure 3: Pass-through ratios for VGG16 (a) for all activation layers and ResNet-18 (b) for activation layers after skip connection tested on Tiny ImageNet dataset for noise-free ($\sigma = 0$) case.

**Pass-through ratio analysis.** Figure 2 compares the distributions of the input and outputs of the activation functions in the SmallCNN trained on the noise-free `CIFAR10` dataset. The SPA layer nullifies fewer features in the first layer while it takes into account the cross-channels feature dependence. The proposed SPA shifts the feature input distribution before sparsifying more important (negative) features (see Figure 2b), which can preserve more information and reduce the information loss caused by the ReLU layer. However, the last activation SPA layer produces more sparse output features, which can adjust the sparsity level of the output features according to the input distribution. This indicates that the SPA layer can improve the network performance and robustness, as well as enhance the network adaptability and flexibility.

Figure 3a compares the pass-through ratios (PTRs) of ReLU and SPA activation functions in the VGG16 network trained on Tiny ImageNet. The results demonstrate how SPA and ReLU differ in their behavior across the network's depth. In the early layers of VGG16, SPA allows more features to

Table 2: Statistical comparison of Accuracy for deep models with ReLU, GELU, and SPA.

| | Noise level $\sigma$ | Activations | | | | | | |
|---|---|---|---|---|---|---|---|---|
| | | ReLU | GELU | | | SPA (ours) | | |
| | | Accuracy, % | Accuracy, % | $\Delta$ | p* | Accuracy, % | $\Delta$ | p* |
| CIFAR100, VGG16 | 0.1 | 56.53 (0.16) | 56.65 (0.26) | 0.12 | 0.2361 | **57.60 (0.17)** | **1.07** | **0.0000** |
| | 0.2 | 47.32 (0.24) | 47.37 (0.18) | 0.05 | 0.5991 | **48.37 (0.13)** | **1.04** | **0.0000** |
| | 0.3 | 40.72 (0.21) | **41.08 (0.24)** | **0.36** | **0.0018** | **41.48 (0.21)** | **0.76** | **0.0000** |
| | 0.4 | 35.94 (0.08) | 36.08 (0.20) | 0.14 | 0.0536 | **36.50 (0.22)** | **0.56** | **0.0000** |
| Tiny ImageNet, ResNet-18 | 0.1 | 42.36 (0.35) | *41.69 (0.35)* | *-0.67* | *0.0007* | **43.62 (0.34)** | **1.25** | **0.0003** |
| | 0.2 | 30.35 (0.58) | *29.58 (0.81)* | *-0.77* | *0.0015* | **31.07 (0.76)** | **0.72** | **0.0016** |
| | 0.3 | 23.72 (0.38) | 23.18 (1.02) | -0.54 | 0.3413 | **25.66 (1.39)** | **1.94** | **0.0238** |
| Caltech256, ResNet-18 | 0.1 | 65.34 (0.47) | *64.57 (0.39)* | *-0.76* | *0.0007* | **66.53 (0.36)** | **1.20** | **0.0000** |
| | 0.2 | 62.33 (0.34) | 62.06 (0.35) | -0.27 | 0.0998 | **63.97 (0.33)** | **1.65** | **0.0000** |
| | 0.3 | 59.98 (0.38) | *59.26 (0.37)* | *-0.72* | *0.0002* | **60.99 (0.33)** | **1.01** | **0.0000** |
| | 0.4 | 57.60 (0.43) | *57.16 (0.30)* | *-0.44* | *0.0126* | **58.68 (0.35)** | **1.08** | **0.0000** |
| Pneumonia-MNIST, ResNet-18 (28) | 0.1 | 89.39 (1.17) | 88.96 (1.69) | -0.43 | 0.5448 | **91.01 (0.56)** | **1.62** | **0.0013** |
| | 0.2 | 84.49 (3.09) | 84.42 (3.32) | -0.06 | 0.9580 | **89.31 (0.57)** | **4.82** | **0.0000** |
| | 0.3 | 80.51 (5.97) | 80.98 (6.36) | 0.46 | 0.8661 | **88.59 (0.77)** | **8.08** | **0.0000** |
| | 0.4 | 80.93 (4.59) | *74.29 (7.02)* | *-6.63* | *0.0237* | **87.72 (3.55)** | **6.79** | **0.0020** |
| DermaMNIST, ResNet-18 (28) | 0.1 | 73.60 (0.43) | 73.78 (0.37) | 0.18 | 0.1560 | **73.94 (0.30)** | **0.34** | **0.0060** |
| | 0.2 | 70.74 (0.55) | 70.78 (0.77) | 0.04 | 0.9101 | **72.16 (0.25)** | **1.42** | **0.0000** |
| | 0.3 | 68.94 (0.66) | 68.84 (0.99) | -0.10 | 0.7909 | **70.81 (0.33)** | **1.87** | **0.0000** |
| | 0.4 | 67.93 (0.92) | 68.07 (0.71) | 0.15 | 0.6926 | **69.77 (0.48)** | **1.85** | **0.0001** |
| OrganAMNIST, ResNet-18 (28) | 0.1[3] | 90.58 (0.50) | 90.47 (0.22) | -0.11 | 0.5414 | **91.14 (0.26)** | **0.57** | **0.0042** |
| | 0.2[3] | 87.93 (0.43) | 87.97 (0.45) | 0.04 | 0.8380 | **89.02 (0.26)** | **1.09** | **0.0003** |
| | 0.3[4] | 85.75 (0.31) | *85.10 (0.67)* | *-0.65* | *0.0109* | **87.68 (0.21)** | **1.93** | **0.0000** |
| | 0.4[4] | 83.61 (0.37) | 83.67 (0.74) | 0.06 | 0.8229 | **85.49 (0.46)** | **1.88** | **0.0000** |

**Note:** * $p < 0.05$ are marked in bold for higher accuracy and in italic for lower accuracy. [2] input image size of $28 \times 28$. [3,4] $\delta$ were initialized from the main and generalized setups based on Bayesian Optimization, accordingly.

Table 3: Accuracy of ResNet-18 with ReLU and SPA activations for ImageNet.

| Median Acc. for SPA, % | Range for SPA | Median Acc. for ReLU, % | Range for ReLU |
|---|---|---|---|
| 66.63% | 66.61%–66.85% | 66.30% | 66.19%–66.57% |

pass through compared to ReLU, suggesting that SPA preserves more key low-level features which are critical to extract higher-level features in deeper layers. This higher pass-through in the initial layers could contribute to better learning and representation of the data. It is crucial to highlight that both activation functions show a decrease in PTR as the network deepens. However, ReLU exhibits a tendency to oversparsify the features due to its simple rectifying operation which allows only nonnegative elements to pass. This leads to a substantial drop in the PTR, with layers 3 to 11 showing a PTR below 50%, and layers 10 and 11 dropping to as low as 20%.

While the intermediate ReLU layers severely sparsifies features, the last ReLU layer reverses this trend by passing more features, with a PTR that is notably higher than 40%. This increase in the pass-through ratio can be seen as a compensatory effect, attempting to rectify the extreme sparsification that occurs in earlier layers. This sudden increase may aim to recover some of the features that were excessively filtered out by previous layers, but it could also indicate inconsistency in feature retention across the network.

SPA, on the other hand, appears to manage this trade-off more effectively by gradually reducing the PTR in deeper layers without abrupt fluctuations, maintaining a more consistent and balanced feature selection process. This controlled sparsification is likely a contributing factor to its higher performance compared to ReLU.

The controlled reduction of PTR in SPA likely encourages better generalization, focusing the network's attention on the most important features. The difference in PTRs is also reflected in the performance, as the SPA activation function achieves a higher accuracy of 52.62%, compared to 50.20% for ReLU. This demonstrates that SPA's dynamic feature passing and sparseness control throughout the network's layers may enhance the network's ability to extract meaningful patterns, particularly in more complex datasets like `Tiny ImageNet`.

Similar behaviors of the SPA layers are observed in ResNet-18, trained on the same Tiny ImageNet dataset as shown in Figure 3b. Both SPA and ReLU tend to reduce the PTR as the network progresses through deeper layers. As with VGG16, SPA in ResNet-18 shows a more controlled and gradual reduction in the PTR, allowing more features to pass in the earlier layers while selectively filtering them as the layers deepen. This leads to a refined representation of features in the later layers.

The PTRs of ReLU in ResNet-18 initially decrease but subsequently increase, suggesting a compensatory mechanism for the extensive sparsification occurring in the earlier layers. This compensatory effect is most pronounced in the final layer, where the PTR becomes markedly higher, as illustrated in 3b. As in VGG16, the final layers of ReLU attempt to rectify this by passing more features, resulting in a pass-through ratio higher than expected in the deeper layers. This fluctuation in ReLU's feature selection process may lead to inefficiencies, which are somewhat mitigated in SPA. SPA's consistent and balanced approach contributes to its slightly better performance, with an accuracy of 54.07% compared to ReLU's 53.15%.

Therefore, in both VGG16 and ResNet-18, SPA demonstrates its ability to manage feature sparsification more effectively, promoting better feature retention and overall network performance.

## 5 DISCUSSION AND CONCLUSION

This study introduced Simplex Projection Activation (SPA), a novel activation function designed to enhance the performance of convolutional neural networks (CNNs) by addressing the limitations of traditional activation functions such as ReLU. Our extensive experimental evaluations across various domain-specific datasets demonstrate SPA's superior classification accuracy. This superiority was consistent across both original and noise-injected data, as confirmed by permutation statistical tests. Our findings suggest that SPA provides a more robust and efficient mechanism for feature activation, thereby enhancing the network's ability to capture complex patterns and improve generalization over traditional methods.

### 5.1 LIMITATIONS AND FUTURE WORK

**Generalization to other architectures**. SPA can be adapted to fully connected architecture by performing along the feature dimension that allows taking into account mutual relations across multiple neurons of linear layers. Besides, the SPA function's ability to project features onto a probability simplex makes it a promising candidate for replacing "softmax" in certain architectures, such as attention mechanisms, to enforce sparsity or enhance interpretability. In other words, the proposed activation could be used in other architectures, like transformers and multilayer perception.

**Computational complexity**. While traditional activation functions like ReLU have $O(n)$ complexity for $n$ elements due to their element-wise nature, SPA introduces an additional computational cost. This projection involves sorting the feature vectors, resulting in a complexity of $O(n \log(n))$, where $n$ is the number of channels in the feature tuple. More information about the computational overhead of SPA is presented in Appendix G). For modern deep networks the additional complexity is manageable, especially in early layers with fewer channels. Thus, developing the faster realizations of SPA is the point of the future research.

**Hyperparameter selection.** As a limitation of the proposed activation layer, the necessity of finding a robust method of defining the parameter $\delta$ and its training setup should be mentioned. This issue becomes relevant for large-scale datasets and models. High computational costs impede the use of iterative parameter search approaches like Bayesian optimization. The proposed method of $\delta$ initialization based on the similarity of the output distribution with ReLU can solve this issue but requires further investigation.

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

## A APPENDIX: SUPPLEMENTARY OUTLINE

This supplementary material describes the experiment setup used in our study, $\delta$ initialization methods, and additional results including various activation functions, such as PReLU He et al. (2015), ELU Clevert et al. (2016), and SELU Klambauer et al. (2017). The outline of the supplementary is presented in Table 4.

Table 4: Supplementary content.

## B  APPENDIX: DATASETS AND PREPROCESSING

A broad range of datasets was used in the experiments, including widely recognized benchmarks: MNIST [LeCun et al. (1998)], FashionMNIST [Xiao et al. (2017)], CIFAR10/100 [Krizhevsky et al. (2009)], Caltech256 [Griffin et al. (2006)], and Tiny ImageNet [Le & Yang (2015)]. Additionally, a smaller subset of the CIFAR10 dataset, comprising 5K training and 5K testing images (referred to as CIFAR10-5K), was created for comparing activation layers on a smaller dataset.

Furthermore, domain-specific datasets were used to assess the applicability of the SPA layer in real-world tasks: GTSRB [Houben et al. (2013)] (traffic signs), SVHN [Netzer et al. (2011)] (street view house numbers), and seven biomedical datasets from the MedMNIST Yang et al. (2023) collection (PathMNIST, PneumoniaMNIST, BreastMNIST, DermaMNIST, OrganAMNIST, OrganCMNIST, and OrganSMNIST).

In addition, experiments with noisy data were performed to evaluate SPA's robustness, an important characteristic for real-world applications. Samples were degraded with random Gaussian noise with zero mean and varying standard deviations ($\sigma$) to simulate noise. The noise was added to the original samples, and the same noisy images were used for training networks with different activation functions. Ten noisy copies of each dataset were generated, with noise levels set at $\sigma = 0.05$, 0.1, 0.2, 0.3, and 0.4. The original dataset ($\sigma = 0$) was also included for comparison. This approach ensures that comparisons between activation functions are based on the same noisy input data.

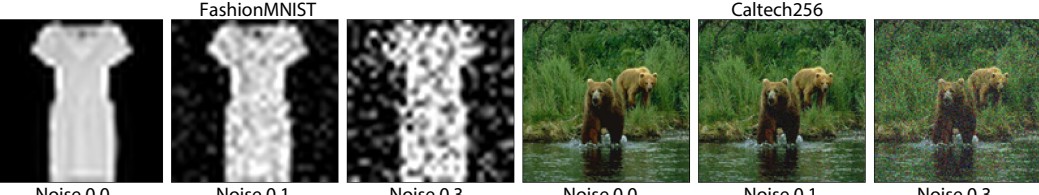

Figure 4: Examples of images in the FashionMNIST and Caltech 256 dataset for different noise levels.

## C  APPENDIX: MODELS AND TRAINING PARAMETERS

### C.1  MODELS

The backbone of the SmallCNN model consists of 3 convolution layer followed by MaxPool, Batch Normalization and activation function. The head part of the SmallCNN consists of one Flatten and one Fully Connected (FC) layer. The SmallCNN architecture is depicted in Fig. 5.

The used VGG16 model consisted of a convolutional neural network (CNN) backbone and a small head consisting of a flatten layer and a fully connected layer. The head was used to transform CNN features (output of the backbone) into predicted classes. The backbone consisted of several stacks of a convolutional layer, batch normalization, and ReLU, separated by maxpool layers with a 2×2 kernel and stride of 2. Each convolutional layer had 3×3 kernel size, padding, and stride equaled 1.

Table 5: Datasets description.

| Dataset | Samples | # classes | image-size | Description |
|---|---|---|---|---|
| MNIST | 70,000 | 10 | 28x28x1 | The classic handwritten digit dataset. |
| FMNIST | 70,000 | 10 | 28x28x1 | The dataset of Zalando's article images. |
| CIFAR10 | 60,000 | 10 | 32x32x3 | The CIFAR-10 is a dataset of 10 classes. |
| CIFAR100 | 60,000 | 100 | 32x32x3 | Like CIFAR-10, but with 100 classes. |
| GTSRB | 52,000 | 43 | WxHx3 | The German Traffic Sign Benchmark. |
| SVHN | 600,000 | 10 | 32x32x3 | The Street View House Numbers. |
| Tiny ImageNet | 100,000 | 200 | 64x64x3 | The tiny version of ImageNet. |
| ImageNet | 14,197,122 | 1,000 | WxHx3 | A large dataset for image recognition. |
| Caltech256 | 30,607 | 257 | 64x64x3 | A superset of the Caltech-101 dataset. |
| PathMNIST | 107,180 | 9 | 28x28x3 | Colon Pathology part of MedMNIST. |
| PneumoniaMNIST | 5,836 | 2 | 28x28x1 | Chest X-Ray part of MedMNIST. |
| BreastMNIST | 870 | 2 | 28x28x1 | Breast Ultrasound part of MedMNIST. |
| DermaMNIST | 10,015 | 7 | 28x28x3 | Dermatoscope part of MedMNIST. |
| OrganAMNIST | 58,830 | 11 | 28x28x1 | Abdominal CT A part of MedMNIST. |
| OrganCMNIST | 23,583 | 11 | 28x28x1 | Abdominal CT C part of MedMNIST. |
| OrganSMNIST | 25,211 | 11 | 28x28x1 | Abdominal CT S part of MedMNIST. |

Table 6: Dataset training/test parameters for SmallCNN, ResNet-18 and VGG16.

| Dataset | Epochs | Train preprocessing | Test preprocessing |
|---|---|---|---|
| MNIST
FMNIST | 200 | resize($32\times32$) | resize($32\times32$) |
| SVHN
GTSRB | 200 | | |
| CIFAR10
CIFAR100
CIFAR10-5K | 350 | resize($32\times32$), RandCrop[1],
RandomFlip, Normalize[2,3,4] | resize($32\times32$),
Normalize[2,3,4] |
| Tiny ImageNet | 300 | | |
| ImageNet | 70 | resize($256\times256$), CenterCrop[1],
resize($224\times224$), Normalize[5,6] | as train
procedure |
| Caltech256 | 300 | | |

**Note:** [1] crop size = $32\times32$, padding = 4; [2] CIFAR10: mean = [0.4914, 0.4822, 0.4465], std = [0.2023, 0.1994, 0.2010]; [3] CIFAR100: mean = [0.5071, 0.4867, 0.4408], std = [0.2675, 0.2565, 0.2761]. [4] Tiny ImageNet: mean = [0.480, 0.448, 0.398], std = [0.272, 0.266, 0.274]. [5] ImageNet: mean = [0.485, 0.456, 0.406], std = [0.229, 0.224, 0.225].

The structure of the VGG16 model with the number of channels for convolutional layers is presented in Table 7.

The ResNet-18 model employed follows a residual network architecture, which uses skip connections to allow gradients to pass through deeper layers more effectively. he network consists of 18 layers, including convolutional layers with $3\times3$ kernels, batch normalization, and ReLU activation. Four residual blocks are employed, each increasing the number of channels (64, 128, 256, and 512) while reducing the spatial resolution. MaxPooling is applied after the first convolutional layer, and the final classification layer is a fully connected layer preceded by a global average pooling layer. Dropout is used in intermediate layers to prevent overfitting.

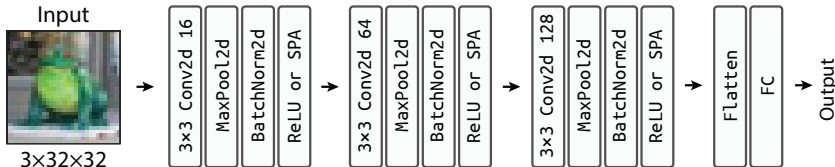

Figure 5: The SmallCNN model architecture. Each convolution layer has $3\times3$ kernel size and padding = 1. Each MaxPool layer was used with $2\times2$ kernel and stride = 2.

Table 7: VGG16 structure. Each conv2d layer is followed by batch normalization and ReLU. Kernel size is $3\times3$, padding and stride equal to 1. Each maxpool layer has $2\times2$ kernel and stride of 2. All blocks are stacked sequentially.

| CNN backbone | | | | | Head |
|---|---|---|---|---|---|
| block1 | block2 | block3 | block4 | block5 | |
| conv2d, 64 | conv2d, 128 | conv2d, 256 | conv2d, 512 | conv2d, 512 | flatten |
| conv2d, 64 | conv2d, 128 | conv2d, 256 | conv2d, 512 | conv2d, 512 | FC |
| maxpool | maxpool | conv2d, 256 | conv2d, 512 | conv2d, 512 | |
| | | maxpool | maxpool | maxpool | |

## C.2 TRAINING PROCEDURE

For SmallCNN, the training procedure was conducted with learning rate: $lr = 0.3 \cdot 10^{-4}$, batch size: 128, Adam optimiser [Kingma & Ba (2015)] with $\beta_1 = 0.9$, $\beta_2 = 0.999$. No learning rate schedule was used. The training procedure of SmallCNN consisted of 200 epochs for `MNIST` and `FashionMNIST` datasets, 350 epochs for `CIFAR10` and `CIFAR100` datasets, and 50 epochs for `GTSRB` and `SVHN`.

For VGG model and `CIFAR10/100` datasets, the same training procedure was applied for 200 epochs.

For `Tiny ImageNet` and `Caltech256` dataset, SGD optimizer with $\mu = 0.9$ and $lr = 0.01$ was used for 300 epochs. In addition, weight decay parameter ($\ell_2$ regularization) of $5 \cdot 10^{-4}$ was set up, which provides higher results together with learning rate schedules. The step learning rate (updated every 100 epochs) and cosine learning rate decreasing ($T = 300$) was used for ResNet-18 and VGG16, accordingly.

The `ImageNet` dataset was tested using an SGD optimizer with $lr = 0.004$ for 70 epochs, with the learning rate decreasing 50 times on the 50th epoch.

For `MedMNIST` datasets, the training parameters followed the setup in Yang et al. (2021): 100 epochs with Adam optimizer, learning rate decreasing by 10 at 50-th and 75-th epochs. The input images of size $28 \times 28$ were converted to RGB format and normalized.

## D APPENDIX: ADDITION RESULTS FOR SMALLCNN

The results for of statistical comparison of Best Accuracy under independent noise versions of SmallCNN with ReLU, GELU, and SPA activations are presented in Table 8.

To gain deeper insights into the performance of various activation functions, we conducted additional experiments by incorporating other commonly used activation functions into our analysis. These experiments aimed to provide a broader comparison and understand how these functions perform under different conditions. We constructed boxplots to assess the accuracy achieved by neural networks utilizing PReLU He et al. (2015), ELU Clevert et al. (2016), and SELU Klambauer et al. (2017) activation functions. These performances were evaluated across different noise levels using the `CIFAR10` and `FashionMNIST` datasets, as shown in Figure 6. The results revealed that, generally, these activation functions, PReLU, ELU, and SELU, tended to yield lower accuracies than ReLU, GELU, and SPA.

Table 8: Statistical comparison of Best Accuracy under independent noise versions of SmallCNN with ReLU, GELU, and SPA activations.

| | Noise level $\sigma$ | Activation | | | | | | |
|---|---|---|---|---|---|---|---|---|
| | | ReLU | GELU | | | SPA (ours) | | |
| | | Accuracy, % | Accuracy, % | $\Delta$ | p* | Accuracy, % | $\Delta$ | p* |
| CIFAR10 | 0.1 | 78.26 (0.14) | **78.45 (0.15)** | **0.19** | **0.0119** | **78.61 (0.14)** | **0.35** | **0.0001** |
| | 0.2 | 71.36 (0.27) | **71.62 (0.10)** | **0.26** | **0.0064** | **71.99 (0.20)** | **0.64** | **0.0000** |
| | 0.3 | 65.61 (0.21) | 65.77 (0.30) | 0.16 | 0.1825 | **66.16 (0.21)** | **0.55** | **0.0000** |
| | 0.4 | 60.64 (0.35) | 60.73 (0.26) | 0.09 | 0.5234 | **61.25 (0.18)** | **0.60** | **0.0003** |
| CIFAR10-5K | 0.1 | 65.11 (0.56) | 65.39 (0.30) | 0.28 | 0.1861 | 65.16 (0.48) | 0.05 | 0.8467 |
| | 0.2 | 59.44 (0.29) | 59.19 (0.45) | -0.25 | 0.1499 | 59.71 (0.53) | 0.27 | 0.1681 |
| | 0.3 | 54.18 (0.38) | 54.23 (0.47) | 0.05 | 0.7925 | **54.80 (0.43)** | **0.62** | **0.0031** |
| | 0.4 | 50.23 (0.34) | 50.07 (0.48) | -0.17 | 0.3800 | **50.89 (0.62)** | **0.66** | **0.0091** |
| CIFAR100[1] | 0.05 | 54.39 (0.24) | 54.41 (0.27) | 0.01 | 0.9103 | 54.34 (0.22) | -0.05 | 0.6193 |
| | 0.1 | 50.32 (0.22) | 50.39 (0.28) | 0.07 | 0.5348 | **50.61 (0.22)** | **0.30** | **0.0090** |
| | 0.2 | 43.51 (0.21) | 43.48 (0.18) | -0.02 | 0.8020 | **43.71 (0.07)** | **0.21** | **0.0085** |
| | 0.3 | 38.01 (0.23) | 38.09 (0.19) | 0.08 | 0.4230 | **38.28 (0.15)** | **0.27** | **0.0067** |

**Note:** * $p < 0.05$ are marked in bold. [1] only first two layers of SmallCNN were replaced for CIFAR100.

More statistical comparisons for `MNIST`, `FashionMNIST`, `GTSRB` and `SVHN` are presented in Table 9. The data in the Table indicates that, on average, the SPA layers tends to converge to local minimum with higher accuracy compared to the ReLU activation function. This observation is statistically significant and consistent across the `MNIST`, `FashionMNIST`, `GTSRB` and `SVHN` datasets, as well as at all levels of noise introduced in the tests.

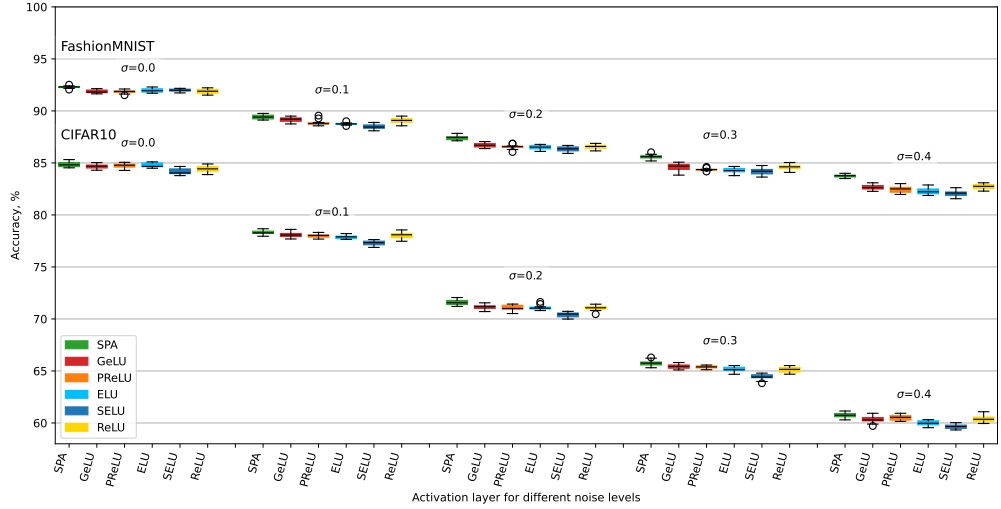

Figure 6: Boxplot analysis of performance of SmallCNN models with different activation layers for the FMNIST and CIFAR10 datasets corrupted at different noise levels $\sigma = \{0, 0.1, 0.2, 0.3, 0.4\}$.

# E    APPENDIX: ADDITION RESULTS FOR DEEP MODELS

This section presents additional results for deep models, such as VGG16 and ResNet-18, on various datasets (e.g., `Tiny ImageNet`, `Caltech 256`, `PathMNIST`, and other medical datasets). The graphs show a comparison of accuracy between SPA, ReLU, and GELU activations for each of the models. As can be seen, for most noise and datasets, SPA activation performs better than ReLU and GELU, confirming its effectiveness in a variety of task conditions.

Table 9: Statistical comparison of Accuracy of SmallCNN with ReLU, GELU, and SPA activations for noisy datasets.

| Noise level | ReLU | GELU | | | SPA (ours) | | |
|---|---|---|---|---|---|---|---|
| | Accuracy, % | Accuracy, % | Δ | p* | Accuracy, % | Δ | p* |
| | | | | MNIST | | | |
| 0.1 | 99.43 (0.03) | 99.42 (0.04) | -0.01 | 0.4218 | **99.51 (0.04)** | **0.08** | **0.0004** |
| 0.2 | 99.23 (0.07) | 99.22 (0.04) | -0.01 | 0.7017 | **99.29 (0.05)** | **0.07** | **0.0257** |
| 0.3 | 98.91 (0.07) | 98.86 (0.07) | -0.05 | 0.1640 | **98.99 (0.05)** | **0.08** | **0.0075** |
| 0.4 | 98.39 (0.13) | 98.36 (0.11) | -0.02 | 0.6551 | **98.55 (0.06)** | **0.16** | **0.0023** |
| | | | | FashionMNIST | | | |
| 0.1 | 89.27 (0.14) | 89.32 (0.13) | 0.04 | 0.4961 | **89.68 (0.10)** | **0.40** | **0.0000** |
| 0.2 | 86.87 (0.25) | 86.95 (0.24) | 0.07 | 0.5119 | **87.52 (0.18)** | **0.64** | **0.0000** |
| 0.3 | 84.98 (0.11) | 84.88 (0.17) | -0.10 | 0.1347 | **85.64 (0.33)** | **0.66** | **0.0000** |
| 0.4 | 82.92 (0.25) | 83.00 (0.17) | 0.08 | 0.4068 | **83.75 (0.22)** | **0.83** | **0.0000** |
| | | | | GTSRB | | | |
| 0.1 | 82.67 (1.05) | 83.06 (0.50) | 0.40 | 0.3011 | **84.94 (0.66)** | **2.28** | **0.0001** |
| 0.2 | 68.88 (0.80) | 69.41 (0.83) | 0.53 | 0.1655 | **72.23 (0.33)** | **3.35** | **0.0000** |
| 0.3 | 58.66 (0.43) | 58.60 (0.67) | -0.06 | 0.8310 | **61.90 (0.73)** | **3.25** | **0.0000** |
| 0.4 | 51.41 (0.72) | 51.62 (0.61) | 0.21 | 0.4920 | **53.80 (0.70)** | **2.39** | **0.0000** |
| | | | | SVHN | | | |
| 0.1 | 85.68 (0.33) | 85.67 (0.22) | -0.01 | 0.9027 | **87.98 (0.11)** | **2.30** | **0.0000** |
| 0.2 | 79.30 (0.17) | *79.13 (0.11)* | *-0.17* | *0.0045* | **81.34 (0.23)** | **2.04** | **0.0000** |
| 0.3 | 72.34 (0.29) | 72.30 (0.26) | -0.05 | 0.6585 | **74.09 (0.25)** | **1.75** | **0.0000** |
| 0.4 | 65.82 (0.40) | *65.48 (0.29)* | *-0.34* | *0.0169* | **67.18 (0.27)** | **1.36** | **0.0000** |

**Note:** * $p < 0.05$ are marked in bold

The comparison for the VGG16 model on the `Tiny ImageNet` dataset can be seen in the Figure 7, and for ResNet-18 on the same dataset in the Figure 8. Also, the results for ResNet-18 on the `Caltech 256` dataset are shown in the Figure 9.

Results for medical dataset (MedMNIST) are presented for `PathMNIST` (Figure 10), `PneumoniaMNIST` (Figure 11), `BreastMNIST` (Figure 12), `DermaMNIST` (Figure 13), `OrganAMNIST` (Figure 14), `OrganCMNIST` (Figure 15), and `OrganSMNIST` (Figure 16). The 3 versions of coefficients are presented the last three datasets. "BO, best" refers to the mean setup in the main text. "BO, average" refers the generalized setup in the main text. Simple, "SPA" refers to the ones multipliers.

The Table 11 shows a comparison of the accuracy of models with different activations (ReLU, GELU and SPA) on data with different noise levels. Testing was carried out on several datasets, including `CIFAR100`, `Tiny ImageNet` and `Caltech 256`, for the VGG16 and ResNet-18 models. This is a more extended table of the one that was in the main text. The table shows the accuracy for different noise levels, as well as the difference (Δ) between the accuracy of GELU and SPA activations compared to ReLU. As can be seen from the table, SPA activation shows better results compared to Rely at almost all noise levels for all datasets.

**Large-scale dataset**. For `ImageNet` dataset, the results for Resnet-18 models are presented in Table 10. Due to the computation power restriction, each trial consisted of 70 epoch.

Table 10: Accuracy of with ReLU and SPA activations for ImageNet.

| Activation | Median Accuracy | Range | # trials | Model |
|---|---|---|---|---|
| ReLU | 66.30% | 66.19%–66.57% | 5 | ResNet-18 |
| SPA | 66.74% | 66.61%–66.85% | 4 | |

Table 11: Accuracy of different nets with ReLU, GELU, and SPA activations for noise data.

| Noise level | Activations | | | | | | |
|---|---|---|---|---|---|---|---|
| | ReLU | GELU | | | SPA | | |
| | Acc., % | Acc., % | $\Delta$ | p* | Acc., % | $\Delta$ | p* |
| | | CIFAR100, VGG16 | | | | | |
| 0.0 | 67.56 (0.17) | **68.07 (0.25)** | **0.51** | **0.0000** | **68.29 (0.18)** | **0.73** | **0.0000** |
| 0.1 | 56.53 (0.16) | 56.65 (0.26) | 0.12 | 0.2361 | **57.60 (0.17)** | **1.07** | **0.0000** |
| 0.2 | 47.32 (0.24) | 47.37 (0.18) | 0.05 | 0.5991 | **48.37 (0.13)** | **1.04** | **0.0000** |
| 0.3 | 40.72 (0.21) | **41.08 (0.24)** | **0.36** | **0.0018** | **41.48 (0.21)** | **0.76** | **0.0000** |
| 0.4 | 35.94 (0.08) | 36.08 (0.20) | 0.14 | 0.0536 | **36.50 (0.22)** | **0.56** | **0.0000** |
| | | Tiny ImageNet (cos) [1,2], VGG16 | | | | | |
| 0.0 | 50.00 (0.39) | **51.88 (0.38)** | **1.88** | **0.0000** | **52.66 (0.27)** | **2.66** | **0.0000** |
| 0.05 | 47.78 (0.28) | **49.47 (0.30)** | **1.68** | **0.0002** | **49.53 (0.19)** | **1.74** | **0.0008** |
| 0.1 | 42.17 (0.28) | *41.70 (0.47)* | *-0.48* | *0.0118* | 41.89 (0.36) | -0.29 | 0.0650 |
| 0.2 | 31.18 (0.34) | *27.71 (0.75)* | *-3.46* | *0.0002* | *29.27 (0.67)* | *-1.91* | *0.0002* |
| 0.3 | 24.13 (0.54) | *20.86 (0.77)* | *-3.27* | *0.0002* | *23.10 (0.47)* | *-1.03* | *0.0023* |
| | | Tiny ImageNet (step) [3,4], ResNet-18 | | | | | |
| 0.0 | 53.01 (0.35) | **53.49 (0.48)** | **0.48** | **0.0156** | **54.31 (0.26)** | **1.30** | **0.0003** |
| 0.05 | 49.84 (0.31) | 49.94 (0.27) | 0.10 | 0.5197 | **50.78 (0.33)** | **0.93** | **0.0016** |
| 0.1 | 42.36 (0.35) | *41.69 (0.35)* | *-0.67* | *0.0007* | **43.62 (0.34)** | **1.25** | **0.0003** |
| 0.2 | 30.35 (0.58) | *29.58 (0.81)* | *-0.77* | *0.0015* | **31.07 (0.76)** | **0.72** | **0.0016** |
| 0.3 | 23.72 (0.38) | 23.18 (1.02) | -0.54 | 0.3413 | **25.66 (1.39)** | **1.94** | **0.0238** |
| | | Caltech 256 [3,5], ResNet-18[6] | | | | | |
| 0.0 | 67.70 (0.31) | *66.72 (0.18)* | *-0.98* | *0.0000* | **68.95 (0.38)** | **1.25** | **0.0000** |
| 0.1 | 65.34 (0.47) | *64.57 (0.39)* | *-0.76* | *0.0007* | **66.53 (0.36)** | **1.20** | **0.0000** |
| 0.2 | 62.33 (0.34) | 62.06 (0.35) | -0.27 | 0.0998 | **63.97 (0.33)** | **1.65** | **0.0000** |
| 0.3 | 59.98 (0.38) | *59.26 (0.37)* | *-0.72* | *0.0002* | **60.99 (0.33)** | **1.01** | **0.0000** |
| 0.4 | 57.60 (0.43) | *57.16 (0.30)* | *-0.44* | *0.0126* | **58.68 (0.35)** | **1.08** | **0.0000** |

**Note:** * $p < 0.05$ are marked in bold for higher accuracy and in italic for lower accuracy

[1] $\delta$ were trained with smaller weight decay in 0.3 times than other model parameters

[2] cosine learning rate schedular was used for training

[3] $\delta$ were trained with smaller weight decay in 0.1 times than other model parameters

[4] step learning rate schedular was used for training

[5] $\delta$ were trained with smaller weight decay in 0.08 times than other model parameters

[6] Base version of ResNet-18 was used (as for ImageNet (input image of $224 \times 224$))

# F   APPENDIX: $\delta$ PARAMETER INITIALIZATION

## F.1   PARAMETER SEARCH FOR SMALLCNN

Figure 17a illustrates the average and peak accuracy levels achieved by the SmallCNN model when employing SPA layers initialized with various $\delta$ values. These experiments were carried out on the CIFAR10 dataset without any added noise. For each $\delta$ setting, five separate trials were conducted. The value of $\delta = 20$ yielded the highest accuracy, along with a robust average accuracy, making it the preferred choice for the base initialization in subsequent experiments. It is important to highlight that the selected value $\delta$ represents the lower limit of the initialization range, with $\delta_{init}$ being uniformly distributed between $\delta$ and $\delta + 1$, denoted as $\delta_{init} \sim \mathcal{U}[\delta, \delta + 1]$.

The evolution of the $\delta$ parameter initialized with $\delta = 20$ for the SmallCNN with CIFAR10 dataset is presented in Figure 17a. One can observe that $\delta$ values did not converge in 350 epochs. In addition, the graph line of the third layer is notably different from the graphs of the first two layers, which implies that the diverse $\delta$ for different layers can be beneficial.

To estimate the influence of the learnability of the $\delta$ parameter and the number of replaced layers, we conducted statistical comparisons for four different SPA setups presented in Table 12. The comparison with the ReLU layer is presented in Table 13. The comparison between SPA (fixed) and SPA is presented in Table 14. The boxplot representation of the comparisons is presented in Figure 18.

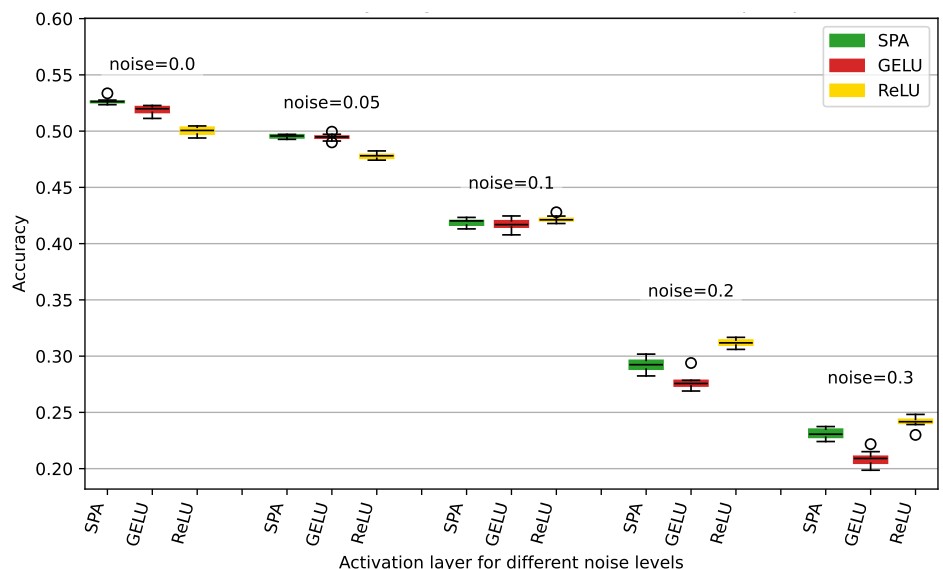

Figure 7: Accuracies for VGG16 for SPA, ReLU and GELU for Tiny ImageNet.

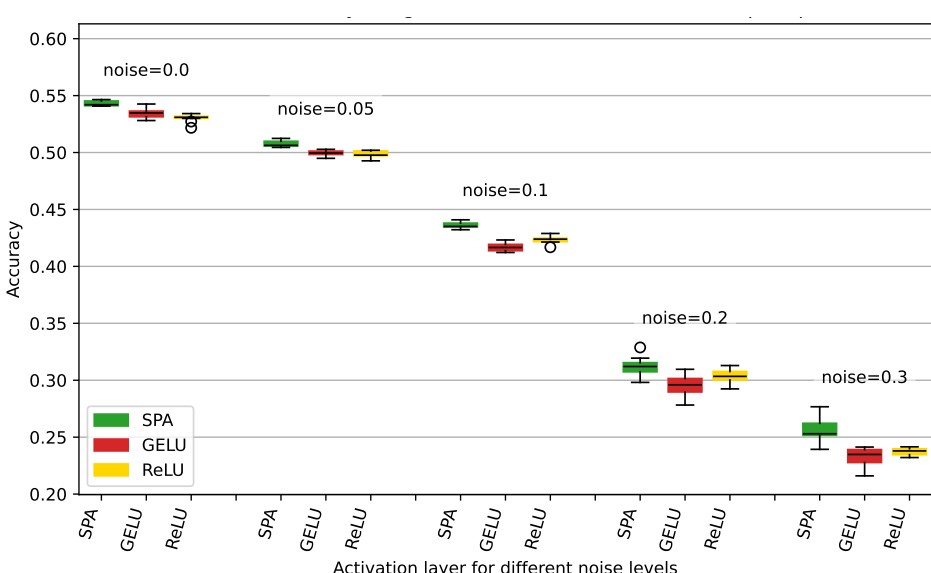

Figure 8: Accuracies for ResNet-18 for SPA, ReLU and GELU for Tiny ImageNet.

From Table 14 we can conclude that there is no statistically significant difference between choosing the best parameter and fixing it or leaving it trainable. According to our experiments from Table 13, when we replace all ReLU layers with SPA the results are better than when we replace only first two layers.

## F.2 PARAMETER SEARCH FOR VGG16

Our evaluation of the VGG16 deep learning model on the CIFAR10 dataset revealed that the initialization of the $\delta$ parameter significantly influences the model's final accuracy. To optimize the initial $\delta$ setting, we employed various strategies:

- Layer-Specific Initialization (Setup 1): We assigned unique $\delta$ values to each layer.

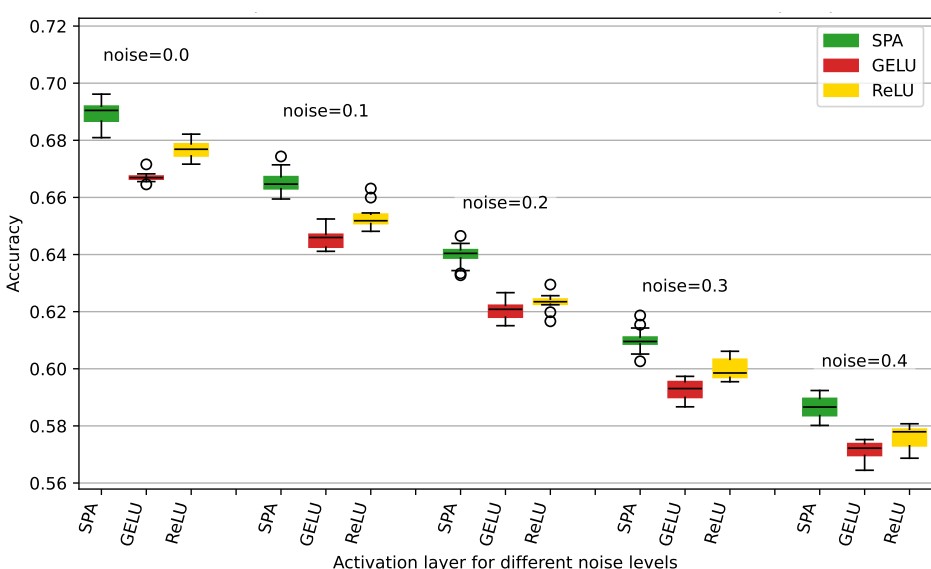

Figure 9: Accuracies for ResNet-18 for SPA, ReLU and GELU for Caltech dataset.

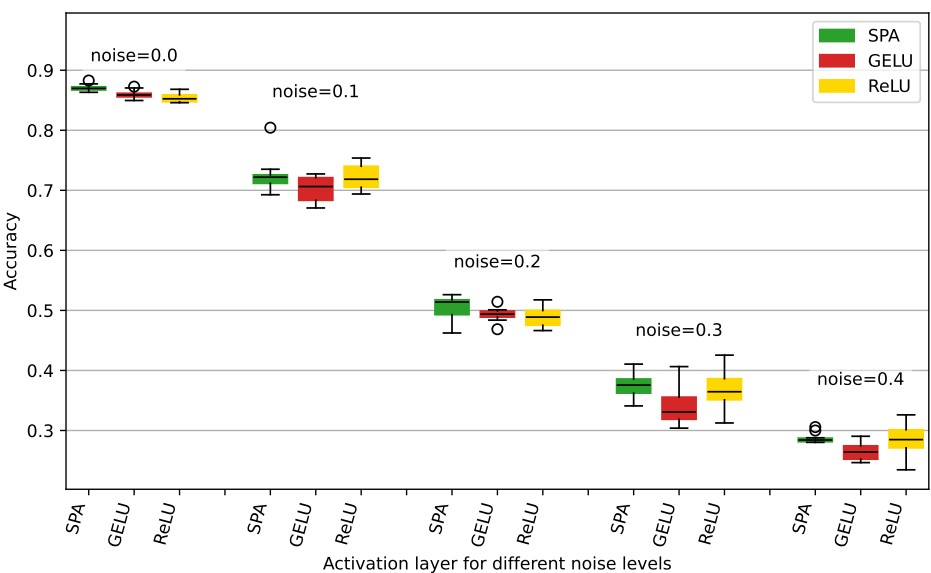

Figure 10: Accuracies for ResNet-18 for SPA, ReLU and GELU for PathMNIST.

Table 12: Different SPA setups. The $\delta$ was initialized from uniform distribution. Replaced ReLU layers: with other activations, SmallCNN has 3 ReLU layers.

| Label | Learnable | $\delta_{init}$ | Replaced layers |
|---|---|---|---|
| SPA (fixed) SPA (fixed, 2 layers) | No | $\mathcal{U}[20, 21]$ | all 1$^{st}$ and 2$^{nd}$ |
| SPA SPA (2 layers) | Yes | $\mathcal{U}[20, 21]$ | all 1$^{st}$ and 2$^{nd}$ |

- Trained Parameter Initialization (Setup 2): We initialized the SPA layers using the final trained $\delta$ values from a previous run and retrained the model.

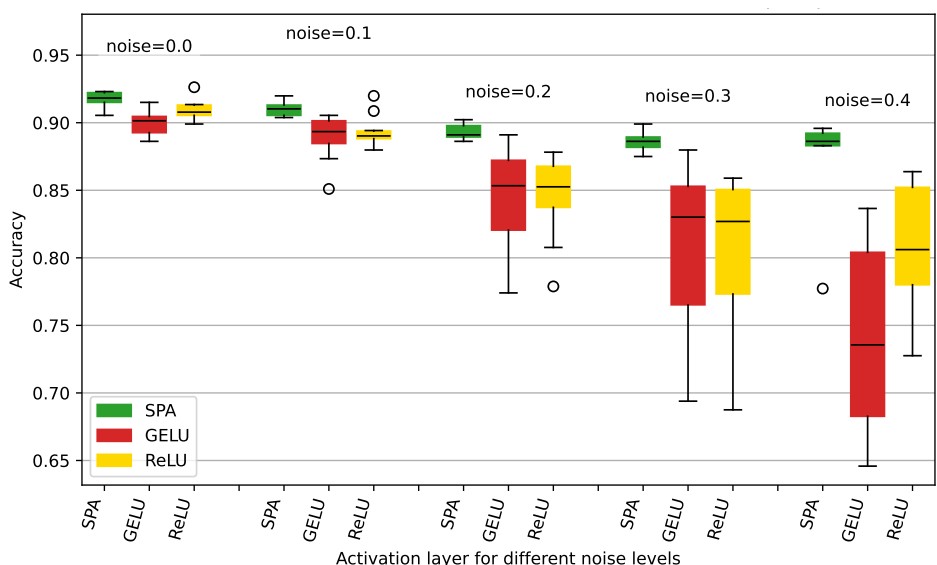

Figure 11: Accuracies for ResNet-18 for SPA, ReLU and GELU for PneumoniaMNIST.

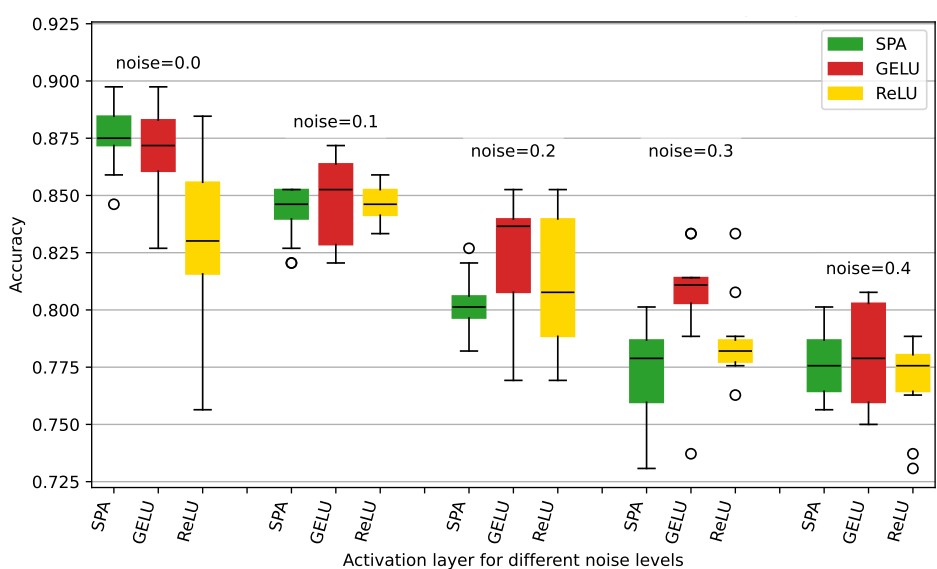

Figure 12: Accuracies for ResNet-18 for SPA, ReLU and GELU for BreastMNIST.

- Uniform Initialization (Identical): We tested a uniform $\delta$ value across all layers
- Selective SPA Layer Initialization (2 layers): We replaced only the first two ReLU layers with SPA layers, maintaining the same initial $\delta$.

The $\delta$ values for these initialization methods are detailed in Table 15, and the corresponding accuracy distributions are illustrated in Figure 20. All experiments were performed on noise-free CIFAR10 data.

### F.3 DELTA INITIALIZATION BASED ON THE SIMILARITY BETWEEN THE EFFECTS OF THE SPA AND RELU ON OUTPUT DISTRIBUTION

For the relatively big datasets and deep neural networks, we suggest using the approach based on the similarity between the effects of the SPA and ReLU on the output distribution. For the normalized

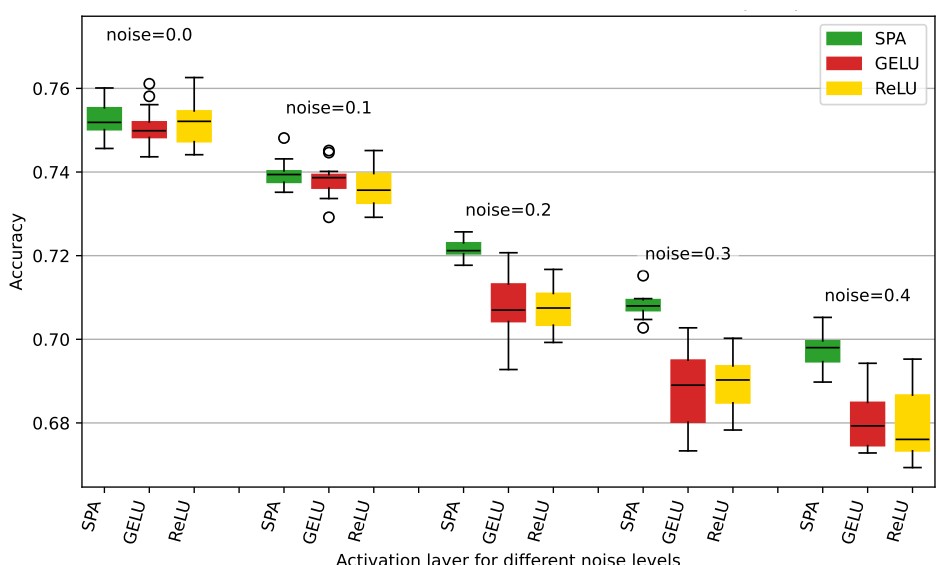

Figure 13: Accuracies for ResNet-18 for SPA, ReLU and GELU for DermaMNIST.

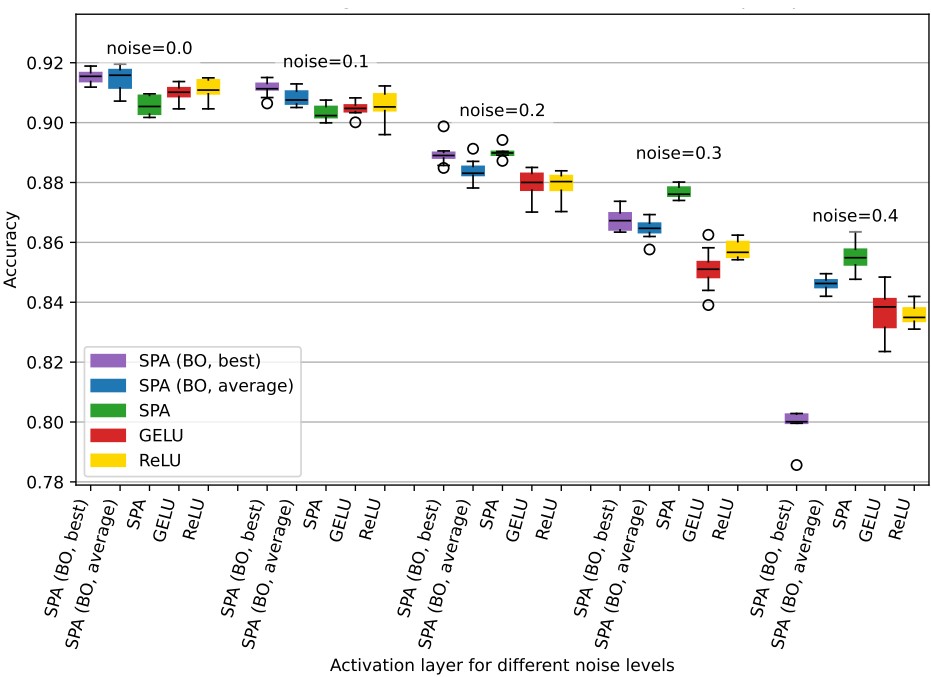

Figure 14: Accuracies for ResNet-18 for SPA, ReLU and GELU for OrganAMNIST.

Gaussian input distribution $\mathcal{N}(0,1)$, we can initialize $\delta$ as $0.4 \cdot C$, where $C$ is a number of channels. In this case, the form and mean value of the output distribution and pass-through ratio for SPA and ReLU actvations will be similar. This method means that for the $\delta$ will be the same for the same number of channels. The coefficient 0.4 is the mean value of output distribution of the ReLU.

## F.4 DELTA INITIALIZATION FROM BAYESIAN OPTIMIZATION

The used ResNet-18 module includes 17 activation layers; however, there are only 4 numbers of channels: 64, 128, 256, and 512. For this number of channels, the delta was initially set based on

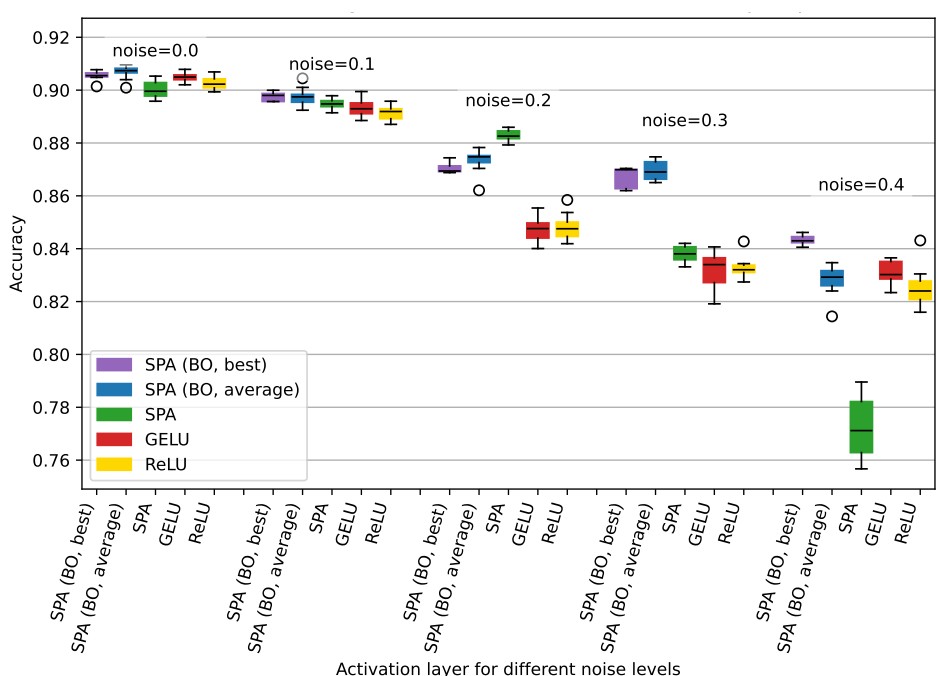

Figure 15: Accuracies for ResNet-18 for SPA, ReLU and GELU for OrganCMNIST.

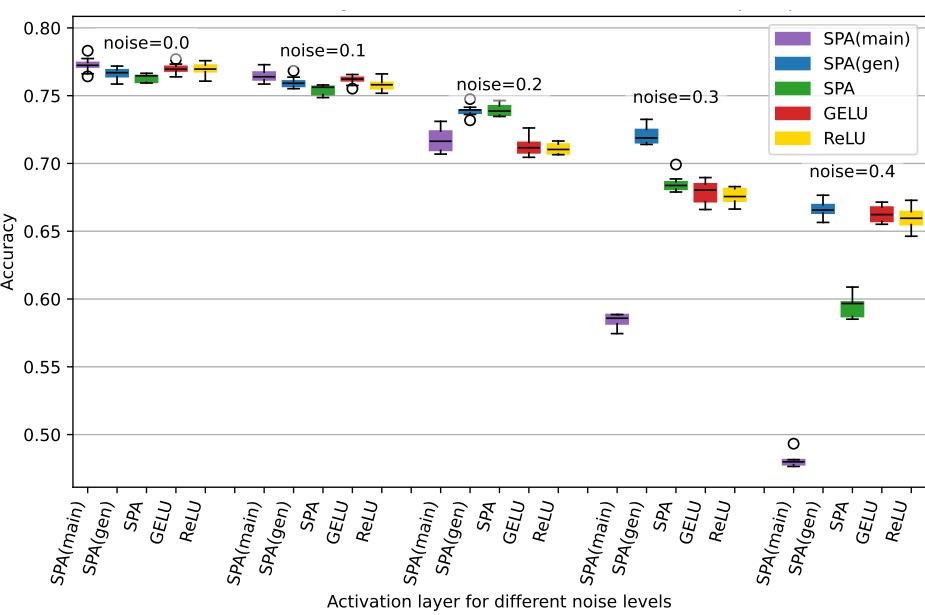

Figure 16: Accuracies for ResNet-18 for SPA, ReLU and GELU for OrganSMNIST.

the similarity between the effects of the SPA and ReLU on output distribution. The 5 or 6 layers of different multipliers were considered for Bayesian optimization search: for the very first activation layer, for 4 ResNet blocks (layers), the ResNet-18, and the very last activations as the 6-th multiplier. The search range was set from 0.1 to 6 with step 0.1.

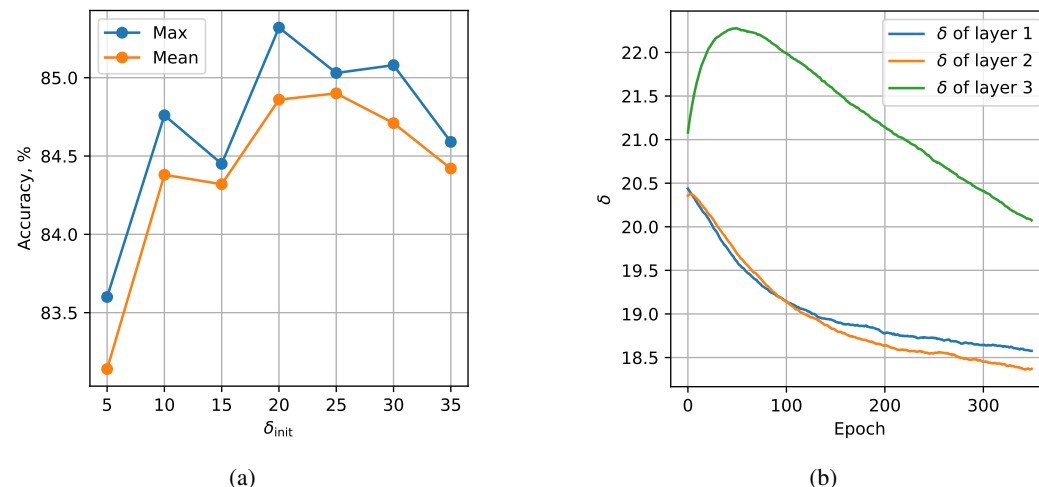

(a)                                                (b)

Figure 17: $\delta$ value selection. (a) Maximum and mean accuracy over 5 trials for SmallCNN with SPA layers with different $\delta$. The presented $\delta$ values are the low bound of the uniform distribution (b) Evolution of $\delta$ parameter during training process.

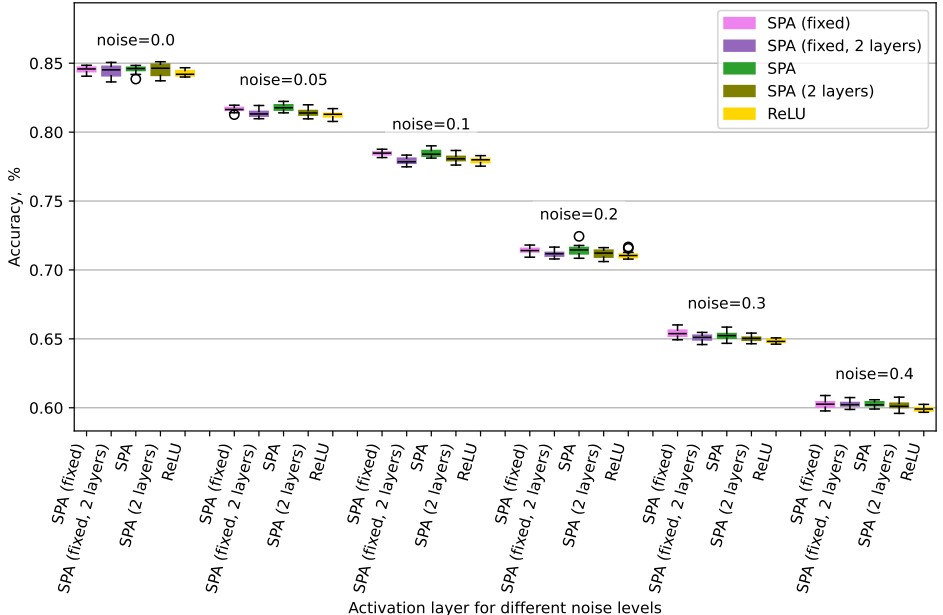

Figure 18: Boxplot of comparison of SPA layer with different $\delta$ parameters for different noise levels and for CIFAR10 dataset.

# G  APPENDIX: COMPUTATIONAL AND TIME COMPLEXITY OF THE SPA LAYER

## G.1  COMPUTATIONAL COMPLEXITY

Let's consider one cross-channel vector in a feature map with size $C$. A ReLU activation will process this vector element-wise with $O(C)$ computational complexity. For SPA, we need to clamp the shift input according to 9. The shifting parameter $\nu*$ can be calculated according to 7, where the set $\mathcal{I}$ is the largest $I$ elements $\{x_1, \dots, x_I\}$ such that the smallest element $x_I > \bar{x}_{\mathcal{I}} - \frac{\delta}{I}$ or satisfying the condition 8. All the mentioned calculations can be done element-wise with $O(C)$ with the already

Table 13: Comparison of SPA layer with different $\delta$ parameters with respect to ReLU for different noise levels and for CIFAR10 dataset.

| Activations | Noise | mean (std) | $\Delta$ | p* | t | n1/n2 |
|---|---|---|---|---|---|---|
| ReLU (base) | 0.0 | 84.27 (0.22) | 0.000 | - | - | 20/20 |
| | 0.05 | 81.25 (0.23) | 0.000 | - | - | 20/20 |
| | 0.1 | 77.95 (0.22) | 0.000 | - | - | 20/20 |
| | 0.2 | 71.07 (0.23) | 0.000 | - | - | 20/20 |
| | 0.3 | 64.84 (0.14) | 0.000 | - | - | 20/20 |
| | 0.4 | 59.90 (0.17) | 0.000 | - | - | 20/20 |
| SPA (fixed) | **0.0** | **84.54 (0.23)** | **0.270** | **0.0008** | **3.77** | **20/20** |
| | **0.05** | **81.66 (0.18)** | **0.410** | **0.0000** | **6.36** | **20/20** |
| | **0.1** | **78.46 (0.17)** | **0.510** | **0.0000** | **8.17** | **20/20** |
| | **0.2** | **71.41 (0.24)** | **0.340** | **0.0001** | **4.63** | **20/20** |
| | **0.3** | **65.40 (0.30)** | **0.560** | **0.0000** | **7.60** | **20/20** |
| | **0.4** | **60.27 (0.28)** | **0.370** | **0.0000** | **4.97** | **20/20** |
| SPA (fixed, 2 layers) | 0.0 | 84.43 (0.42) | 0.160 | 0.0552 | 1.97 | 20/20 |
| | 0.05 | 81.35 (0.27) | 0.100 | 0.1957 | 1.32 | 20/20 |
| | 0.1 | 77.91 (0.23) | -0.040 | 0.6022 | -0.52 | 20/20 |
| | 0.2 | 71.18 (0.22) | 0.110 | 0.1476 | 1.48 | 20/20 |
| | **0.3** | **65.09 (0.24)** | **0.250** | **0.0003** | **4.10** | **20/20** |
| | **0.4** | **60.25 (0.21)** | **0.350** | **0.0000** | **5.67** | **20/20** |
| SPA | **0.0** | **84.57 (0.19)** | **0.300** | **0.0000** | **5.19** | **20/20** |
| | **0.05** | **81.77 (0.25)** | **0.520** | **0.0000** | **6.95** | **20/20** |
| | **0.1** | **78.46 (0.25)** | **0.510** | **0.0000** | **6.82** | **20/20** |
| | **0.2** | **71.43 (0.37)** | **0.360** | **0.0005** | **3.68** | **20/20** |
| | **0.3** | **65.23 (0.32)** | **0.390** | **0.0000** | **5.02** | **20/20** |
| | **0.4** | **60.25 (0.21)** | **0.350** | **0.0000** | **5.72** | **20/20** |
| SPA (2 layers) | **0.0** | **84.54 (0.45)** | **0.270** | **0.0034** | **3.18** | **20/20** |
| | 0.05 | 81.40 (0.27) | 0.150 | 0.0589 | 1.95 | 20/20 |
| | 0.1 | 78.11 (0.29) | 0.160 | 0.0505 | 2.02 | 20/20 |
| | 0.2 | 71.21 (0.28) | 0.130 | 0.1124 | 1.62 | 20/20 |
| | **0.3** | **65.03 (0.22)** | **0.190** | **0.0020** | **3.29** | **20/20** |
| | **0.4** | **60.19 (0.30)** | **0.290** | **0.0004** | **3.80** | **20/20** |

Table 14: Comparison of SPA layer with fixed and learnable parameters with respect to each other.

| Noise level | Mean (std) of accuracy, % | | Stat. parameters | | | |
|---|---|---|---|---|---|---|
| | SPA | SPA (fixed) | $\Delta$ | p* | t | n1/n2 |
| 0.0 | 84.57 (0.19) | 84.54 (0.23) | -0.030 | 0.5918 | -0.55 | 20/20 |
| 0.05 | 81.77 (0.25) | 81.66 (0.18) | -0.110 | 0.1033 | -1.67 | 20/20 |
| 0.1 | 78.46 (0.25) | 78.46 (0.17) | 0.000 | 0.9916 | 0.01 | 20/20 |
| 0.2 | 71.43 (0.37) | 71.41 (0.24) | -0.020 | 0.8696 | -0.17 | 20/20 |
| 0.3 | 65.23 (0.32) | 65.40 (0.30) | 0.170 | 0.0943 | 1.73 | 20/20 |
| 0.4 | 60.25 (0.21) | 60.27 (0.28) | 0.020 | 0.8035 | 0.24 | 20/20 |

defined set $\mathcal{I}$. The implemented algorithm of set $\mathcal{I}$ identification consists of sorting, cumulative sum, and several simple element-wise operations. All of them have $O(C)$ complexity except the sorting operation. The fastest sorting algorithms have computation complexity $O(n \log(n))$, which is the speed bottleneck compared to other operations. Therefore, the implemented SPA layer has $O(C \log(C))$ computation complexity.

It should be noted that to identify set $\mathcal{I}$ we need to find only the minimum element of $x_I$. Then, we can define the full set $\mathcal{I}$ by selecting all the elements of the input that are greater than or equal to

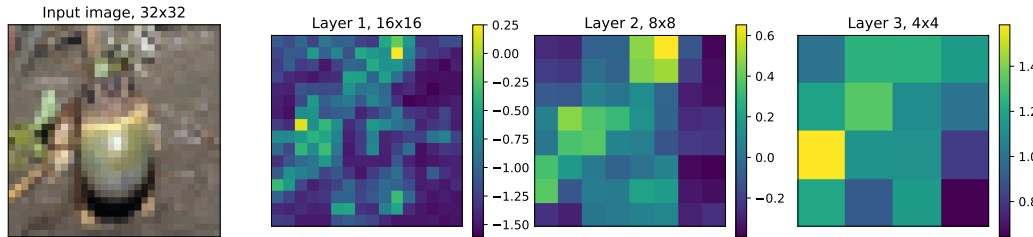

Figure 19: Shifting values ($\nu^\star$) of the update rule (9) for SmallCNN on noise-free CIFAR10 dataset.

Table 15: The $\delta$ values for different initialization methods for VGG16. The initialization of $\delta$ is: $\delta_{init} \sim \mathcal{U}[\delta', \delta' + 1]$.

|   | Convolutional Layers | $\delta'$ | | | |
|---|---|---|---|---|---|
|   |   | Identical | 2 layers | Setup 1 | Setup 2 |
| 1 | block1.conv2d.64 | 20 | 20 | 20 | 20 |
| 2 | block1.conv2d.64 | 20 | 20 | 20 | 22 |
| 3 | block2.conv2d.128 | 20 | - | 30 | 33 |
| 4 | block2.conv2d.128 | 20 | - | 30 | 34 |
| 5 | block3.conv2d.256 | 20 | - | 50 | 57 |
| 6 | block3.conv2d.256 | 20 | - | 50 | 58 |
| 7 | block3.conv2d.256 | 20 | - | 50 | 59 |
| 8 | block4.conv2d.512 | 20 | - | 80 | 90 |
| 9 | block4.conv2d.512 | 20 | - | 80 | 86 |
| 10 | block4.conv2d.512 | 20 | - | 80 | 83 |
| 11 | block5.conv2d.512 | 20 | - | 80 | 80 |
| 12 | block5.conv2d.512 | 20 | - | 80 | 80 |
| 13 | block5.conv2d.512 | 20 | - | 80 | 83 |

$x_I$. Hence, the sorting of all elements is excessive, and faster implementations of a SPA layer are possible.

The conclusions above are related to one tuple along the channels dimension with a size of $C$. The full computational complexity of the feature map with a size of $B \times C \times H \times W$ is $O(BHWClog(C))$ for SPA and $O(BHWC)$ for ReLU.

### G.2 TIME COMPLEXITY EXPERIMENTS

To test the real time consumption, we have collected the evaluation and training time for one epoch for ResNet-18 on the Tiny-ImageNet dataset. The time complexity tests were conducted for the GPU NVIDIA RTX 2080 SUPER with CUDA 11.1, Python 3.9.13, PyTorch 1.13.1, and 2x CPU: AMD EPYC 7352 24-Core Processor. The results are presented in Table 16. Deep models (ResNet-18 and VGG-16) with SPA activations showed approximately 3 times more evaluation time and 2 times more training time than with ReLU activation functions. However, SPA tends to find the optimal point faster than ReLU, as can be seen based on the difference in the best epoch. In other words, more computation consumption of the SPA can be compensated for by faster finding of the optimal local minimum for some cases (ResNet-18).

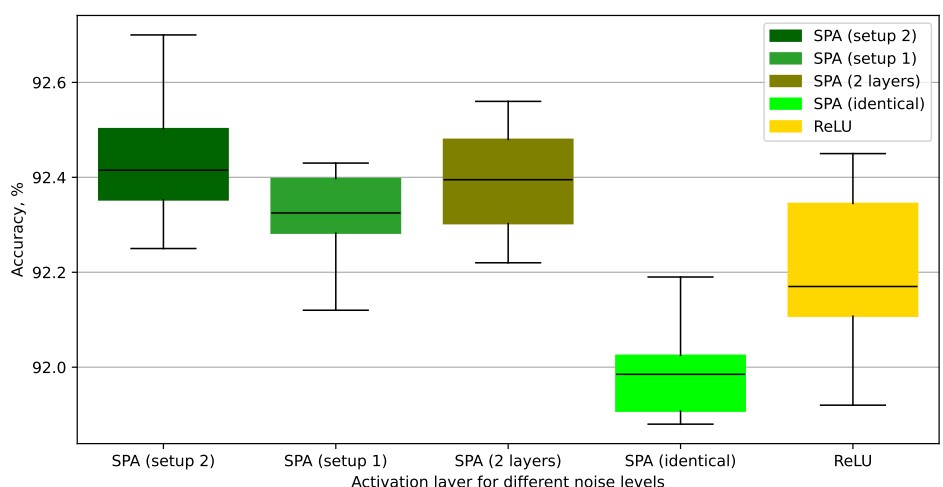

Figure 20: Boxplot of VGG16 model accuracy across various $\delta$ initialization methods on CIFAR10 dataset.

Table 16: Time comparison of ResNet-18 and VGG-16 and SmallCNN with ReLU and SPA activations on Tiny-ImageNet.

| Activation | evaluation time, s | train time, s | best epoch, mean (std) |
|---|---|---|---|
| ResNet-18 | | | |
| ReLU | 2 | 48 | 239.4 (47.2) |
| SPA | 6 | 92 | 104.0 (2.4) |
| VGG16 | | | |
| ReLU | 1 | 26 | 283.5 (6.2) |
| SPA | 3 | 49 | 280 (8.3) |

