# OpenReview forum: "CROSS-CHANNEL ACTIVATION FUNCTION WITH PASS-THROUGH RATIO CONTROL"
_ICLR.cc/2025/Conference — Submitted to ICLR 2025_

### Official Review · Reviewer_vsPb · 2024-11-01

**Soundness:** 3
**Presentation:** 2
**Contribution:** 3
**Rating:** 6
**Confidence:** 4

**Summary:**

In this paper, a new cross-channel activation function, namely SPA, is proposed. Most of previous activation functions used in deep learning, such as ReLU, only independently consider the multi-channel features, which may ignore some cross channel information. This paper considers to interpret activation functions as an optimization problem, then proposes SPA to maintain the feature relationship between multiple channels. Specifically, each cross channel feature x should be projected to a convex set S (which is defined by introducing a constant \delta), and the projected feature can be viewed as the output of SPA. Moreover, this paper provides the solution of the SPA optimization problem, and show the update rule of each x. The experimental results imply that SPA show good perfermance on variety of databases.

**Strengths:**

1. The consideration that transfer the activation functions into an optimization problem supports the SPA method.
2. The SPA activation function is well-defined, and it is possible to be implemented easily.
3. The relationship between the constant \delta and classification perfermance is carefully analyzed, and the authors provide a way to find a suitable \delta.
4. Experimental results show that SPA show slightly better accuracy than traditional activation functions on multiple models and databases.

**Weaknesses:**

1. The experimental results only include the small-scale databases. The authors mentioned that the imagenet-1k results are included in Appendix E, but I cannot find them there. Moreover, I believe that Imagenet-1k results are important for this paper, which should be included in the main paper instead of appendix.

2. It is better to consider the time-cost of SPA. Is it similar with traditional activation functions?

**Questions:**

1. Please provide experimental results on Imagenet-1k database

2. Is SPA has similar time-complexity with traditional activation functions?

---

### Official Review · Reviewer_uEHc · 2024-11-03

**Soundness:** 2
**Presentation:** 2
**Contribution:** 2
**Rating:** 5
**Confidence:** 4

**Summary:**

In this paper, the author proposes a cross-channel activation function. The core concept of this activation function embraces the cross-channel relationship, which is purported to capture the patterns and semantics of the input data for activation. Additionally, the author introduces a threshold V* for activation, which is utilized to eliminate unimportant features with varying control ratios.
The author applies the aforementioned technique to the cross-channel activation function, which is validated on several toy datasets. The enhancement is somewhat restricted.

**Strengths:**

1) The author presents a novel activation function.
2) The proposed methodology is validated across six datasets.

**Weaknesses:**

1) The concept conveyed in the paper lacks significance. The author introduced two methodologies: cross-channel relation for activation and threshold v* to filter out irrelevant features.

1.1) The author posited that "These functions often process inputs separately, neglecting dependence between them, such as the spatial or cross-channel relation of the features. Spatial relation refers to the local connectivity and neighborhood structure of the features, while cross-channel relation refers to the correlation and diversity of the features across different channels. "
In my opinion, the convolutional operation already calculates the cross-channel relation of the features. Therefore, introducing another cross-channel relation for activation function seems superfluous.

1.2) Regarding the threshold v*, feature normalization and bias serve a similar purpose. Consequently, the significance of the threshold appears diminished.

2) The proposed methodology has only been validated on toy datasets and tiny ImageNet. Larger-scale datasets are imperative. In my view, if the model is trained with an adequate number of dataset samples, the original cross-channel relation learned through convolution operation and the threshold will be well assimilated by the model.

3) The in-depth analysis explaining why deep models necessitate additional cross-channel relation and threshold parameters is absent.

4) The literature review is lacking. Several crucial and highly relevant works are absent.

References:
[1] Dynamic Neural Response Tuning, ICLR 2024.
[2] Exploring optimal adaptive activation functions for various task，IEEE BIBM 2020.
[3] Exploring Optimal Adaptive Activation Functions for Various Tasks, 2020.
[4] Deep sparse rectifier neural networks.  JMLR 2011
[5] Density Modeling of Images using a Generalized Normalization Transformation, CoRR 2015.
[6]  ...

**Questions:**

What distinguishes the cross-channel information acquired through the proposed activation from that obtained through the original convolutional operation?

---

### Official Review · Reviewer_qcFG · 2024-11-03

**Soundness:** 4
**Presentation:** 2
**Contribution:** 3
**Rating:** 6
**Confidence:** 3

**Summary:**

The paper introduces Simplex Projection Activation (SPA), a novel activation function for CNNs that addresses the limitations of conventional activation functions like ReLU by considering cross-channel dependencies. SPA projects input tuples across channels onto a convex set, preserving feature relations and avoiding information loss. The authors also explore the learnable parameter δ, which controls the pass-through ratio and significantly influences the model's performance. Through extensive experiments, the authors demonstrate SPA's effectiveness, showing it outperforms ReLU and other activation functions in various datasets and noise conditions.

**Strengths:**

### Originality

The paper presents the Simplex Projection Activation (SPA) function, which introduces a novel approach to activation functions in convolutional neural networks (CNNs). The originality of the paper lies in its consideration of cross-channel dependencies, which traditional activation functions like ReLU ignore. By projecting input tuples across channels onto a convex set, SPA maintains feature relations and avoids information loss, offering a creative solution to a known limitation in neural network design.

### Quality

The quality of the paper is reflected in its thorough experimental evaluation. The authors have conducted extensive experiments across various datasets and noise conditions, demonstrating SPA's superiority over ReLU and other common activation functions. The statistical tests used to compare the accuracy of different activation functions are appropriate, and the results are consistently presented, indicating a high level of quality in the research methodology.

### Clarity

The paper is generally well-structured and clear in its presentation. The problem statement is clearly defined, the motivation for the SPA function is well-articulated, and a relatively complete derivation process of the simplex method is provided. The use of illustrations and charts to help visualize concepts and results further enhances the clarity of the paper.

### Significance

The significance of the paper is evident in its potential impact on the field of deep learning. SPA's ability to improve model performance and robustness to noise is a valuable contribution, especially given the widespread application of CNNs across various domains. The paper's findings could lead to improvements in the design of neural networks and potentially extend to other types of neural network architectures, highlighting the broader implications of the research.

In conclusion, the paper is strong in its original approach to addressing a known issue in CNNs, the quality of its experimental validation, the clarity of its presentation, and the significance of its potential impact on the field of deep learning. The research presented in this paper could influence future work in activation function design and neural network optimization.

**Weaknesses:**

### Mistake in Mathematical Expression

One of the specific weaknesses in the paper is the definition of the set $S$ used in the SPA function. The paper states $S = \{x = [x_1, x_2, \ldots, x_C] \mid |x_1| + |x_2| + \cdots + |x_C| \leq \delta\}$ without explicitly requiring $x \geq 0$. This omission may compromise the theoretical foundation, as the non-negativity constraint is crucial for the simplex projection and the activation function's behavior. The authors should clarify this condition to avoid any misinterpretation.

### Misleading Illustration

The three-dimensional illustration in Figure 1(b) appears to be hand-drawn, with the projection directions of various points and the axes appearing inconsistent and chaotic, which may mislead readers. High-quality and accurate visual representation is crucial for conveying mathematical concepts, and the quality of this figure does not meet this standard. The authors should consider revising this figure using mathematical 3D space plotting tools such as GeoGebra to ensure it accurately represents the SPA projection without misleading readers.

### Theoretical Implications

While the paper provides a thorough experimental evaluation, it could benefit from a deeper theoretical analysis of the SPA function. Specifically, the paper could explore the theoretical implications of the SPA function on network convergence and generalization. A more in-depth theoretical discussion would strengthen the paper's contribution and provide a stronger foundation for the experimental results.

### Generalization to Other Network Architectures

The paper focuses on the application of SPA to CNNs, but does not extensively explore its potential application to other types of neural network architectures, such as transformer models. Expanding the scope of the paper to include experiments or a discussion on the applicability of SPA to these architectures would enhance its significance and impact.

### Discussion on Limitations

The paper could benefit from a more explicit discussion on the limitations of the SPA function. For example, the authors could discuss potential challenges in optimizing the δ parameter for deep networks or the computational overhead introduced by the SPA function. Acknowledging and addressing these limitations would provide a more balanced view of the SPA function's practical applicability.

In summary, the paper's weaknesses can be addressed by clarifying mathematical definitions, improving visual representations, expanding the theoretical analysis, exploring the applicability to other network architectures, and discussing the limitations of the proposed method. By addressing these points, the paper could provide a more comprehensive and robust contribution to the field of neural network activation functions.

**Questions:**

1. **Clarification on Set $S$ Definition:** In the definition of the set $S$, it was noted that the non-negativity constraint $x \geq 0$ was not explicitly stated. Could the authors please clarify whether this constraint is intended to be part of the definition of $S$?
2. **Applicability to Other Network Architectures:** Given the novelty of the SPA function, it would be insightful to understand its potential application beyond CNNs. Are there any empirical results or theoretical predictions regarding SPA's applicability in other neural network architectures such as RNNs or transformer models?
3. **Theoretical Analysis of SPA:** The paper could benefit from a deeper theoretical analysis of the SPA function, particularly regarding network convergence and generalization. Are there any theoretical insights or ongoing work that the authors could share regarding these aspects?

---

### Official Review · Reviewer_Vxhv · 2024-11-03

**Soundness:** 2
**Presentation:** 2
**Contribution:** 2
**Rating:** 6
**Confidence:** 3

**Summary:**

This paper proposes a new activation function for convolutional neural networks (CNNs) called Simplex Projection Activation (SPA).  Unlike activation functions like ReLU treating elements independently, SPA considers the relationships across multiple channels. Designed as a projection to simplex regularized on $l_1$ norms, the proposed method introduces a rather flexible threshold regarding the norm of different channels.

**Strengths:**

* This paper proposes a new simplex projection perspective for a new activation function, which is new and interesting.

* The paper conducts comprehensive experiments on various model architectures and datasets. The paper also tests the robustness of the proposed method against noise.

**Weaknesses:**

* The paper claims that ReLU suffers information loss by eliminating negative features. However, the proposed method also eliminates features. Moreover, the ReLU masks element-wise features while the proposed method masks channel-wise features. It seems the proposed method would suffer more information loss. Therefore, I have doubts about the analysis regarding the shortcomings of ReLU and how the proposed method improve it.

* According to the experimental results, the improvement brought by SPA is marginal. An average accuracy and its standard deviation over multiple runs should be provided.

* Besides, suppose SPA does improve upon ReLU and GELU. The improvement brought by SPA seems not substantial compared to the increase in computational cost for a much more complicated activation function. I would appreciate a more comprehensive time complexity analysis of the proposed SPA.

**Questions:**

I list some of my concerns in the Weaknesses section. Following are my questions and further concerns.

* Regarding the first weakness I mentioned above, activation functions such as Leaky-ReLU do not eliminate negative features.  Is there a comparison between the proposed method and Leaky-ReLU?

* This paper analyzes the pass-through ratio between the proposed SPA and ReLU. Honestly, I can't see a clear pattern indicating SPA is superior. Why would SPA outperform ReLU-like activation functions? In my understanding, the pass-through ratio of the ReLU-like activation function is controlled by the bias term in the convolutional layer or the linear layer, which is learned automatically during the training procedure with gradient descent. The proposed SPA actually adds a manually determined soft threshold $\delta$ with very complicated computation to determine the set $\mathcal{I}$ of unmasked channels. Is there any theoretical analysis to prove that SPA would outperform ReLU?

* In lines 206-209, the authors claim that masking channel-wise feature (as in SPA) is better than masking element-wise feature (as in ReLU). Is there any theoretical or empirical evidence to support the claim? Would a more coarse-grained activation function lead to more information loss? In SPA, the output of some convolutional kernels is completely masked and set to zero.

---

### Meta-Review · Area_Chair_U5DJ · 2024-12-21

**Metareview:**

This paper proposes a method to consider cross-channel dependency in CNN activation functions. The proposed method shows better performance than typical CNN activation functions like ReLU. In initial reviews, reviewers raised concerns about limited experimental results, justification of the proposed method, and lack of analysis on additional computation costs. The authors provided thorough responses in their rebuttal, and the final score is 6,6,6,5.

While the rebuttal addressed many concerns, the limited experiments (conducted only on small-scale experiments) make this a borderline paper rather than a clear acceptance. Particularly, when proposing improvements to foundational functions of neural networks, it's crucial to demonstrate generalization ability across various backbone types, model scales, target tasks, and data domains. Even considering computational constraints, these concerns remain unaddressed. Given these limitations and ICLR's competitive nature, the AC recommends rejection.

**Additional Comments On Reviewer Discussion:**

While the reviewers found the paper to be an interesting one and were satisfied with the rebuttal, they provided borderline accept/reject recommendations due to remaining concerns about the experimental scale and validation of practicality. The AC agrees with the reviewers' assessment and encourages the authors to submit to a future conference after addressing these aspects.

---

### Decision · Program_Chairs · 2025-01-22

Reject